# Glad: A Streaming Scene Generator for Autonomous Driving

**Bin Xie**[1*†], **Yingfei Liu**[2†], **Tiancai Wang**[2], **Jiale Cao**[1‡], **Xiangyu Zhang**[2,3]
[1]Tianjin University, [2]MEGVII Technology, [3]StepFun
`{bin_xie,connor}@tju.edu.cn`
`{liuyingfei,wangtiancai,zhangxiangyu}@megvii.com`

## Abstract

The generation and simulation of diverse real-world scenes have significant application value in the field of autonomous driving, especially for the corner cases. Recently, researchers have explored employing neural radiance fields or diffusion models to generate novel views or synthetic data under driving scenes. However, these approaches suffer from unseen scenes or restricted video length, thus lacking sufficient adaptability for data generation and simulation. To address these issues, we propose a simple yet effective framework, named Glad, to generate video data in a frame-by-frame style. To ensure the temporal consistency of synthetic video, we introduce a latent variable propagation module, which views the latent features of previous frame as noise prior and injects it into the latent features of current frame. In addition, we design a streaming data sampler to orderly sample the original image in a video clip at continuous iterations. Given the reference frame, our Glad can be viewed as a streaming simulator by generating the videos for specific scenes. Extensive experiments are performed on the widely-used nuScenes dataset. Experimental results demonstrate that our proposed Glad achieves promising performance, serving as a strong baseline for online video generation. We will release the source code and models publicly.

## 1 Introduction

Autonomous driving tasks are usually data-intensive, relying on vast amounts of data to learn informed models. In recent years, autonomous driving has achieved considerable progress, particularly in the field of Bird's Eye View (BEV) perception (Huang et al., 2021; Liu et al., 2022; Li et al., 2022b; Liu et al., 2023; Li et al., 2023b; Liao et al., 2022) and end-to-end planning (Shi et al., 2016; Jiang et al., 2023; Chen et al., 2024), thanks to the availability of public datasets such as nuScenes (Caesar et al., 2019), CARLA (Dosovitskiy et al., 2017), Waymo (Ettinger et al., 2021), and ONCE (Mao et al., 2021). However, these real-world driving datasets exist several limitations. On one hand, it is expensive and labor-intensive to collect large-scale real-world driving data. On the other hand, although corner-cases comprise only a small portion of dataset, they are more important for evaluating the safety of autonomous driving. Therefore, the scale and diversity of real-world datasets constrain the further development of autonomous driving.

Instead of collecting large-scale real-world datasets, researchers have explored generating synthetic street-view data. Some approaches focus on leveraging Neural Radiance Fields (NeRF) (Yang et al., 2023a; Yan et al., 2024a; Yang et al., 2023c; Tonderski et al., 2023) and 3D Gaussian Splatting (3DGS) (Yan et al., 2024b) for rendering the novel views in driving scenes. These cutting-edge approaches offer a powerful tool for rendering highly realistic and detailed images from various viewpoints. However, as shown in Fig. 1(a), these approaches struggle to reconstruct the non-seen streets when the source trajectory and simulated trajectory are different. Recently, some approaches (Gao et al., 2023; Yang et al., 2023b; Swerdlow et al., 2024; Li et al., 2023a; Jia et al., 2023; Hu et al., 2023) attempt to employ Stable Diffusion (Rombach et al., 2022) to generate the synthetic data of driving scenes. For instance, Panacea (Wen et al., 2023) and DriveDreamer-2 (Zhao et al., 2024)

---

*This work was done during the internship at MEGVII Technology
†Equal contribution
‡Corresponding author: Jiale Cao

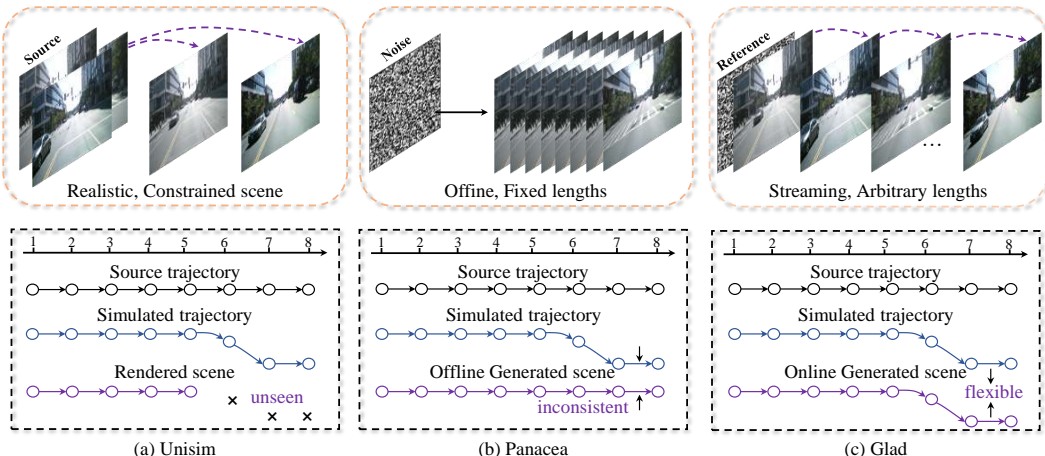

Figure 1: **Comparison of Unisim, Panacea, and our proposed Glad:** (a) The Nerf-based Unisim (Yang et al., 2023c) struggles to render unseen scene and objects when the simulated ego trajectory deviates from the source data. (b) The diffusion-based Panacea (Wen et al., 2023) generates fixed-length video data in an offline manner. It suffers from relatively high memory consumption, and lacks the adaptability to accommodate variations in dynamic simulated trajectory. (c) Our Glad is designed for fame-by-frame generation, enabling to generate videos of arbitrary lengths and exhibiting good flexibility in the variations of simulated trajectory.

have delved into the ability of Stable Diffusion to generate synthetic video data from a given initial frame. However, as shown in Fig. 1(b), these diffusion models (e.g., Panacea) are limited to offline video generation and require high memory consumption. Moreover, it fails to dynamically generate video data in response to the changes in simulated trajectory. For a flexible generator, it requires the ability of generating dynamic scenes corresponding to the simulated trajectory.

To address the issues mentioned above, we propose Glad, a simple yet effective framework that generates and forecasts synthetic video data in an online (i.e., frame-by-frame) manner. Our Glad is based on Stable Diffusion, and introduces the latent variable propagation (LVP) to maintain temporal consistency when performing frame-by-frame generation. In LVP, the latent features of previous frame are viewed as the noise prior injected into the latent features of current frame. In addition, we introduce a streaming data sampler (SDS), which aims to keep the consistency between training and inference, and present an efficient data sampling for training. In SDS, we sample original frames in video clip one by one at continuous iterations, and save the generated latent features in cache used for the noise prior of next iteration.

With such streaming generation manner, our Glad is capable of generating videos of arbitrary lengths in theory. Specifically, it can generate the videos of the novel scene from the noise when serving as the data generator. In addition, given the reference frame, it can produce the video data of specific scene. We perform the experiments on the public autonomous driving dataset nuScenes, which demonstrates the efficacy of our Glad. The contributions and merits can be summarized as follows:

- A simple yet effective framework, named Glad, is proposed to generate video data in an online manner, instead of offline fixed-length manner. Theoretically, our proposed Glad is able to generate videos of arbitrary lengths, showcasing substantial potential in data generation and simulation.

- A latent variable propagation strategy is introduced to ensure the temporal consistency for frame-by-frame video generation, where the latent features of previous frame are viewed as the noise fed to the latent features of current frame.

- We further design a streaming data sampler to provide efficient training. In the streaming data sampler, we sample the frame in video clip one by one at several continuous iterations, and save the latent features in cache for next iteration.

- The experiments are performed on the widely-used dataset nuScenes. Our Glad is able to generate high-quality video data as generator. Further, our generated video data can significantly improve the perception, tracking, and HD map construction performance.

## 2 RELATED WORKS

### 2.1 VIDEO GENERATION MODELS

Recent years have seen considerable advancements in video generation, focusing on improving the quality, diversity, and controllability of generated content. Initial methods (Yu et al., 2022; Clark et al., 2019; Tulyakov et al., 2018) extend the success of Generative Adversarial Networks (GANs) from image synthesis to video generation. However, these approaches often face challenges in maintaining temporal consistency.

The introduction of diffusion models (Ho et al., 2020; Rombach et al., 2022) represents a paradigm shift, providing a robust alternative for high-fidelity video synthesis. VDM (Ho et al., 2022b) utilizes factorized spatio-temporal attention blocks to generate 16-frame, 64×64 pixels videos, which are then upscaleable to $128 \times 128$ pixels and 64 frames using an enhanced model. ImagenVideo (Ho et al., 2022a) further progresses the field through a cascaded diffusion process, beginning with a base model that produces videos of 40×24 pixels and 16 frames, and sequentially upsampling through six additional diffusion models to reach $1280 \times 768$ pixels and 128 frames. MagicVideo (Zhou et al., 2022) employs Latent Diffusion Models (LDM) to enhance efficiency in video generation. VideoLDM (Blattmann et al., 2023), leveraging a similar architecture, incorporates temporal attention layers into a pre-trained text-to-image diffusion model, excelling in text-to-video synthesis. TRIP (Zhang et al., 2024) proposes a new recipe of image-to-video diffusion paradigm that pivots on image noise prior derived from static image. In contrast, our Glad uses latent features from the previous frame as noise prior, replacing Gaussian noise, while TRIP predicts a "reference noise" for the entire clip, risking inaccuracy over time. Vista (Gao et al., 2024) presents a generalizable driving world model with high fidelity and versatile controllability. Additionally, Sora (Tim et al., 2024) significantly enhances video quality and diversity, while introducing capabilities for text-prompt driven control of video generation.

Despite these advancements, video generation still faces several challenges, including maintaining temporal consistency, generating longer videos, and reducing computational costs. A streaming video generation pipeline could present an elegant solution. Notably, several advanced works (Yan et al., 2021; Hong et al., 2022; Huang et al., 2022; Henschel et al., 2024) have adopted an autoregressive approach to generate video frames. VideoGPT (Yan et al., 2021) predicts the current latent code from the previous frame, Autoregressive GAN (Huang et al., 2022) generates frames based on a single static frame, and CogVideo (Hong et al., 2022) employs a GPT-like transformer architecture. ART·V (Weng et al., 2024) firstly explores autoregressive text-to-video generation and adopts multi-reference frames and anchor frame as condition, while our Glad achieves streaming video generation by using previous latents as initial noise in the diffusion process. Nonetheless, these methods do not yet fully satisfy the stringent requirements for generation quality, controllability, and motion dynamics essential in autonomous driving applications.

### 2.2 DRIVING DIFFUSION MODELS

In the realm of driving scenario generation, early research predominantly focuses on synthesizing individual images. BEVGen (Yan et al., 2024b) explores the generation of multi-view street images from a BEV layout. BEVControl (Yang et al., 2023b) incorporates cross-view attention to enhance visual consistency and ensure a cohesive scene representation. MagicDrive (Gao et al., 2023) highlights the challenges associated with the loss of 3D geometric information after the projection from 3D to 2D. In addition to the cross-view consistency, cross-frame consistency remains crucial for temporal modeling. Based on this point, DrivingDiffusion (Li et al., 2023a) and Panacea (Wen et al., 2023) introduce sequences of BEV layouts to generate complex urban videos in a controllable manner. However, their sliding-window approach to video generation proves inefficient and unsuitable for extended durations. Moreover, DriveDreamer-2 (Zhao et al., 2024) and ADriver-I (Jia et al., 2023) use the initial frame as a reference, demonstrating significant potential in autonomous driving simulation by predicting subsequent video sceness. GAIA-I (Hu et al., 2023) generates long-duration, realistic video data conditioned on diverse inputs such as images, text, and action signals. Unfortunately, the capability to manipulate motion trajectories of other vehicles through control signals is limited in (Hu et al., 2023). To meet the demands of data generation and simulation, the challenges such as extended video length, controllability, and consistency still hold significant value.

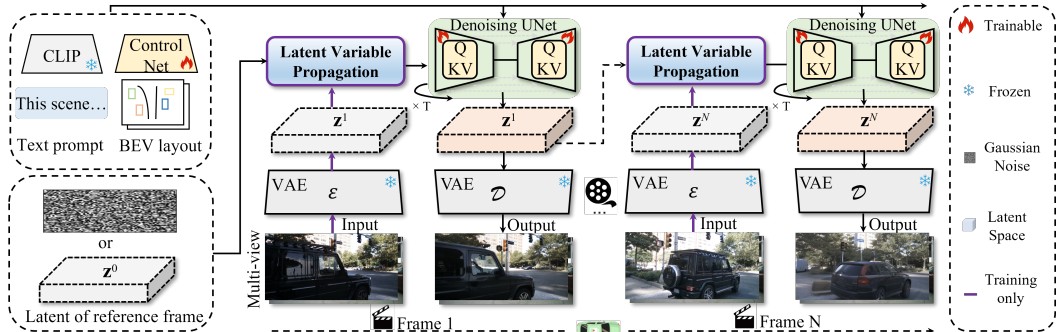

Figure 2: **Overall architecture of our proposed Glad.** Glad is based on Stable Diffusion, and can take random noise or a reference frame as input to generate new or specific scenes. Afterwards, Glad generates video sequences from frame 1 to frame N orderly. We employ the proposed latent variable propagation module to feed the denoised latent features at previous frame to current frame as the noise prior, which can maintain video temporal consistency. This frame-by-frame generation strategy enables to generate videos of arbitrary lengths. In addition, we employ ControlNet (Zhang et al., 2023) to introduce BEV layout for fine-grained control on data generation.

## 3 METHOD

Here we introduce our proposed Glad, a simple yet effective approach that denoises the current frame from the fully denoised latent feature of the previous frame instead of Gaussian noise, which generates and simulates video data in an online manner. The proposed Glad is based on Stable Diffusion (Rombach et al., 2022), and introduces two novel modules for generating video data, including latent variable propagation and streaming data sampler. The latent variable propagation aims to maintain temporal consistency when performing frame-by-frame generation, while streaming data sampler provides an efficient data sampling pipeline and keeps data consistency between training and inference. Our proposed Glad is capable of both generating an entirely new scene from Gaussian noise and simulating a specific scene based on a given reference frame.

**Overview.** Fig. 2 presents the overall architecture of our proposed Glad. Given the Gaussian noise or latent features $z^0$ of the reference frame, we introduce latent variable propagation (LVP) to process it first and then view it as the noise-added latent features $z^1$ of frame 1. Afterwards, we employ denoising UNet to recover the latent features $z^1$, which are fed to VAE decoder for image generation and LVP module for latent features propagation. By analogy, we can generate synthetic images from frame 1 to frame $N$, which together form a continuous video sequence. Compared to the fixed-length video generation in existing approaches (Wen et al., 2023; Zhao et al., 2024), our frame-by-frame video generation approach is more efficient and flexible, which can theoretically generate the videos of arbitrary lengths. In addition, similar to most existing approaches, we employ ControlNet (Zhang et al., 2023) and CLIP (Radford et al., 2021) to condition feature extraction of denoising UNet by given text prompt and BEV semantic layout.

### 3.1 LATENT VARIABLE PROPAGATION

Our latent variable propagation (LVP) aims to maintain the temporal consistency in frame-by-frame video generation. The distribution of noise plays an essential role in conditioning image synthesis of diffusion models. When extending image-level diffusion model Stable Diffusion (Rombach et al., 2022) to generate video sequences, how to generate the noise distribution across time is crucial to maintain video temporal consistency. One straightforward way is to sample the noise from the same distribution for image generation at different frames similar to offline approaches (Wen et al., 2023). However it cannot maintain temporal consistency in our frame-by-frame generation design. To address this issue, our proposed LVP views the denoised latent features at previous frame as the noise prior for image generation at current frame. Based on the LVP module, we can orderly generate video sequences from frame 1 to frame $N$, while maintaining good temporal consistency.

Specifically, we perform the forward diffusion process of frame $n$ based on the latent features of frame $n-1$ as

$$q(\mathbf{z}_{1:T}^n|\mathbf{z}_0^n) = \prod_{t=1}^T q(\mathbf{z}_t^n|\mathbf{z}_{t-1}^n), \ q(\mathbf{z}_t^n|\mathbf{z}_{t-1}^n) = \mathcal{N}(\mathbf{z}_t^n; \sqrt{1-\beta_t}\mathbf{z}_{t-1}^n, \beta_t\varphi(\mathbf{z}_0^{n-1})), \ n=1,...,N \quad (1)$$

where $\mathbf{z}_t^n$ represents the noisy latent features at frame $n$ at time-step $t$, and $\mathbf{z}^{n-1} = \mathbf{z}_0^{n-1}$ represents the denoised latent features at frame $n-1$ generated by denoising UNet. The function $\varphi$ aims to normalize the latent features $\mathbf{z}^{n-1}$ as the noise, where we adopt layer normalization operation. The reverse process at frame $n$ can be written as

$$p_\theta(\mathbf{z}_{0:T}^n) = p(\mathbf{z}_T^n) \prod_{t=1}^T p_\theta(\mathbf{z}_{t-1}^n|\mathbf{z}_t^n), \ p_\theta(\mathbf{z}_{t-1}^n|\mathbf{z}_t^n) = \mathcal{N}(\mathbf{z}_{t-1}^n; \boldsymbol{\mu}_\theta(\mathbf{z}_t^n, t), \boldsymbol{\Sigma}_\theta(\mathbf{z}_t^n, t)), n=1,...,N$$

$$(2)$$

where $p(\mathbf{z}_T^n) = p(\mathbf{z}_0^{n-1})$ is the Gaussian noise for generator or latent features of reference frame for simulator when $n=1$, and represents the denoised latent features $\mathbf{z}^{n-1}$ of generated image at frame $n-1$ when $n=2,...,N$.

### 3.2 STREAMING DATA SAMPLER

As mentioned above, our proposed Glad requires the denoised latent features $\mathbf{z}^{n-1}$ at frame $n-1$ as noise prior to generate the image at frame $n$. To ensure the consistency between training and inference, we need to calculate the denoised latent features $\mathbf{z}^{n-1}$ at frame $n-1$ when performing image generation at frame $n$ during training. However, it is time-consuming and unnecessary to generate the denoised latent features $\mathbf{z}^{n-1}$ at frame $n-1$ from scratch (*i.e.,* frame 1). To address this issue, we introduce a streaming data sampler that samples video clip orderly at continuous iterations, which can improve the efficiency of training.

Fig. 3 presents the pipeline of our streaming data sampler module. Given a video clip having $N$ frames for each clip, we orderly sample the frame one by one at $N$ continuous iterations. At

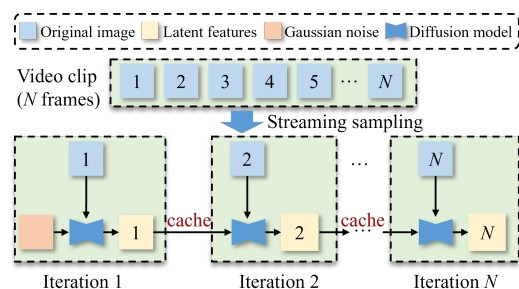

Figure 3: **Illustration of streaming data sampler.** The streaming data sampler samples video clip from frame 1 to frame $N$ at continuous iterations. At each iteration, we save the denoised latent features generated by diffusion model in the cache, and reuse it as the noise at next iteration.

each iteration, we employ diffusion model to generate denoised latent features at current frame, and save the denoised latent features at current frame in the cache. The saved denoised latent features at current iteration will be used as the noise prior at next iteration. In this streaming way, we just generate the synthetic image of each frame in video clip only once, resulting in more efficient training.

### 3.3 TRAINING AND INFERENCE STRATEGY

**Layout and text control.** It is crucial to freely manipulate the agents within the scenes when performing image generation. Following existing works (Wen et al., 2023; Gao et al., 2023), we employ BEV layout and text prompt to precisely control the scene composition, which includes 10 common object categories and 3 different types of map elements. For BEV layout, we firstly convert them into camera view perspective and extract the control elements as object bounding boxes, object depth maps, road maps, and camera pose embeddings. Afterwards, we integrate them into denoising UNet using the ControlNet (Zhang et al., 2023) framework. The text prompts are fed to denoising UNet via pre-trained CLIP (Radford et al., 2021).

**Training.** Inspired by Panacea (Wen et al., 2023), we adopt a two-stage training strategy and employ the same data and processing rules. First, we perform image-level pre-training. Our Glad can also be seen as an image synthesis model, where we only employ Stable Diffusion architecture to generate

the image of one frame from Gaussian noise. Second, we perform video-level fine-tuning on the autonomous driving dataset nuScenes by using our proposed streaming data sampler. To obtain images with a resolution of $512\times256$ pixels, we randomly resize the original images to a proportion of 0.3 to 0.36, and then perform a random crop. This ensures consistency in processing with the downstream StreamPETR method.

**Inference.** Our Glad can be used as generator and simulator, which respectively generate video sequence from Gaussian noise and reference frame. Specifically, given the reference frame or Gaussian noise, we first employ VAE encoder to generate the latent features. Subsequently, we feed it to the latent variable propagation module as noise prior, and employ denoising UNet and VAE decoder to predict the denoised latent features and generate the image at frame 1. Then, the denoised latent features at frame 1 are fed to the latent variable propagation module for image generation at next frame. In this way, our Glad is able to generate a video sequence of arbitrary lengths.

## 4 EXPERIMENTS

### 4.1 DATASETS AND IMPLEMENTATION DETAILS

**nuScenes dataset.** The nuScenes dataset was collected from 1000 different driving scenes in Boston and Singapore. These scenes are split into training, validation, and test sets. Specifically, the training set contains 700 scenes, the validation set contains 150 scenes, and the test set contains 150 scenes. The dataset has totally 10 object classes, and provides accurate 3D bounding-box annotations for each object. In every scene, there are 6 camera views and each view records a length of about 20 second driving video.

**Evaluation metrics.** We evaluate the quality of the generation and simulation, respectively. To quantitatively evalute the generation quality, we adopt the frame-wise Fréchet Inception Distance (FID) (Parmar et al., 2022) and Fréchet Video Distance (FVD) (Unterthiner et al., 2018), which calculate the Fréchet distance between generated content (image or video) and ground truth. To evaluate the model's ability to adhere the given input conditions, such as the BEV layout, we follow (Liao et al., 2022) and report the performance of online vectorized HD map construction. This includes the average precision (AP) for constructing pedestrian crossings ($AP_{ped}$), lane dividers ($AP_{divider}$), road boundaries ($AP_{boundary}$), and their weighted average (mAP). To evaluate the potentiality of Glad as simulator, we assess the 3D detection performance on the nuScenes dataset, including nuScenes Detection Score (NDS) and mean Average Precision (mAP). To further evaluate the temporal consistency of the generated videos, we also report the multi-object tracking results, including average multiple object tracking accuracy (AMOTA) and average multiple object tracking precision (AMOTP).

**Implementation details.** Our Glad is implemented based on Stable Diffusion 2.1 (Rombach et al., 2022). We train our models on 8 NVIDIA A100 GPUs with the mini-batch of 2 images. During training, we first perform image-level pre-training. Constant learning rate $4\times10^{-5}$ has been adopted, and there are 1.25M iterations totally. Afterwards, we fine-tune our Glad on nuScenes dataset with same settings for 48 epochs. We split each video into 2 clips to balance video length and data diversity. During inference, we utilize the DDIM (Song et al., 2020) sampler with 25 sampling steps and scale of the CFG as 5.0. The image is generated at a spatial resolution of $256 \times 3072$ pixels with 6 different views, and split it to 6 images of $256 \times 512$ pixels for evaluation. We adopt StreamPETR (Wang et al., 2023a) with ResNet-50 (He et al., 2016) backbone as perception model. Regarding the map construction model, we employ MapTR (Liao et al., 2022) with ResNet-50 and retrain it under a $512 \times 256$ pixels resolution setting. The inference time of complete denoising process is reported in single NVIDIA A100 GPU.

### 4.2 MAIN RESULTS

**Generation quality.** We first online generate video data on nuScenes validation set, and employ the sliding window strategy to evaluation synthetic video quality similar to existing approach (Wen et al., 2023). Tab. 1 compares data generation quality of our proposed Glad and some state-of-the-art approaches. We present the results of our Glad respectively using Gaussian noise or from a given reference frame. When generating video using Gaussian noise, Glad achieves an FID of 12.57 and an FVD of 207. Furthermore, when generating videos starting from reference frame, Glad achieves a

Table 1: **Comparison of video generation with state-of-the-art approaches** in terms of FID and FVD metrics on nuScenes validation set. Compared to the Single frame approaches, our online approach Glad supports high-quality multi-frame generation. Compared to the offline approaches, our Glad achieves promising performance in an online manner.

| Type | Method | Ref-Frame | Multi-View | Multi-Frame | FID↓ | FVD↓ |
|---|---|---|---|---|---|---|
| Single | BEVGen (Swerdlow et al., 2024) | | ✓ | | 25.54 | N/A |
| | BEVControl (Yang et al., 2023b) | | ✓ | | 24.85 | N/A |
| | MagicDrive (Gao et al., 2023) | | ✓ | | 16.20 | N/A |
| Offline | DriveDreamer (Wang et al., 2023b) | ✓ | | ✓ | 52.60 | 452 |
| | WoVoGen (Lu et al., 2023) | ✓ | | ✓ | 27.60 | 418 |
| | Panacea (Wen et al., 2023) | | ✓ | ✓ | 16.96 | 139 |
| | MagicDrive (Gao et al., 2023) | | ✓ | ✓ | 16.20 | 218 |
| | DrivingDiffusion (Li et al., 2023a) | | ✓ | ✓ | 15.83 | 332 |
| | Drive-WM (Wang et al., 2024) | | ✓ | ✓ | 15.80 | 123 |
| Online | Glad (Ours) | | ✓ | ✓ | **12.57** | **207** |
| | Glad (Ours) | ✓ | ✓ | ✓ | **11.18** | **188** |

lower FID of 11.18 and FVD of 188. Compared to the offline approaches, our online Glad achieves promising performance in terms of both FID and FVD. For instance, our Glad without reference frame outperforms MagicDrive by 3.63 and 11 in terms of FID and FVD respectively. We evaluate the generation speed of our method and some open-source methods. To generate multi-view frames, BEVGen requires 6.6s, MagicDrive takes 11.2s, and our Glad takes 9.4s. Namely, our Glad has a comparable generation speed with these approaches, but has better generation quality.

Table 2: **Performance on downstream tasks of the generated data** on the validation set. We employ a pre-trained perception model StreamPETR to perform 3D detection and multi-object tracking on generated and real data. We report the results of StreamPETR when applied to the real nuScenes validation set as Oracle. The input image size is set as $512\times256$ pixels.

| Method | nuImage | Video Length | mAP↑ | NDS↑ | AMOTA↑ | AMOTP↓ |
|---|---|---|---|---|---|---|
| Oracle | | ≤41 | 34.5 | 46.9 | 30.2 | 1.384 |
| Oracle | ✓ | ≤41 | 37.8 | 49.4 | 33.9 | 1.325 |
| Panacea (Wen et al., 2023) | | 8 | 18.8 | 32.1 | - | - |
| Panacea (Wen et al., 2023) | ✓ | 8 | 19.9 | 32.3 | - | - |
| DriveWM (Wang et al., 2024) | ✓ | 8 | 20.7 | - | - | - |
| | ✓ | 1 | 26.8 | 40.0 | 19.7 | 1.563 |
| Glad (Ours) | | 8 | 26.3 | 39.6 | 20.0 | 1.563 |
| | ✓ | 8 | **28.3** | **41.3** | **22.7** | **1.526** |
| | ✓ | 16 | 27.7 | 40.5 | 21.9 | 1.535 |

**Practicality for generation.** To validate the practicality of our method in driving scenes, we generate all frames in the nuScenes validation set using BEV layout. Instead of using sliding window strategy, we generate the video per 8 frames. We employ the video-based 3D object detection approach, StreamPETR (Wang et al., 2023a), to perform 3D object detection on both ground-truth and synthetic validation set. Tab. 2 reports 3D detection and multi-object tracking results. It can be observed that, compared to Panacea (Wen et al., 2023), our Glad performs better on all 3D object detection metrics, which reflect that our Glad can generate more realistic videos for 3D object detection. Further, our Glad achieves an mAP of 28.3 and an NDS of 41.3, which correspond to 74.9% and 83.6% of the Oracle, respectively. We also report the results when generating the video per 16 frames, the performance of the 3D object detection and tracking drop slightly.

Another crucial application is the expansion of driving dataset to boost the performance of perception model. We first generate synthetic nuScenes training set using Gaussian noise as the reference frame and BEV layout. Then, we first pre-train StreamPETR on this synthetic training set, and then fine-tune it on real training set. Tab. 3 presents the results of baseline, Panacea (Wen et al., 2023), and our Glad. Compared to the offline Panacea, our online approach Glad achieves the comparable performance, significantly outperforming the baseline. We further validate the effectiveness using nuImage pre-training that can provide a better initialization. The baseline achieves the NDS of 49.4 and AMOTA of 33.9, having 2.5 and 3.7 improvements compared to using ImageNet (Deng et al., 2009) pre-training, respectively. Compared to the stronger baseline, our Glad also has 1.9 improvement in terms of NDS, and 2.6 improvement in terms of AMOTA.

Table 3: **Impact of training augmentation using generated synthetic data**. In top part, we employ generated data to pre-train the perception model StreamPETR, and then train it on real data. We set the official reported results where StreamPETR trains on real data only as Baseline. In bottom part, we orderly train the StreamPETR on nuImage, generated data, and real data.

| Method | nuImage | Generated | Real | mAP↑ | NDS↑ | AMOTA↑ | AMOTP↓ |
|---|---|---|---|---|---|---|---|
| Baseline | | | ✓ | 34.5 | 46.9 | 30.2 | 1.384 |
| Panacea | | ✓ | ✓ | 37.1 (+2.6) | 49.2 (+2.3) | 33.7 (+3.5) | 1.353 |
| Glad (Ours) | | ✓ | ✓ | 37.1 (+2.6) | 49.2 (+2.3) | 33.4 (+3.2) | 1.356 |
| Baseline | ✓ | | ✓ | 37.8 | 49.4 | 33.9 | 1.325 |
| Glad (Ours) | ✓ | ✓ | ✓ | 39.8 (+2.0) | 51.3 (+1.9) | 36.5 (+2.6) | 1.295 |

Table 4: **Performance on downstream tasks of the online vectorized HD map construction.** We retrain the MapTR-tiny under 512×256 pixels with ResNet-50 backbone, and report the results on real data as Oracle or Baseline.

| | Method | Video Length | $AP_{ped}$↑ | $AP_{divider}$↑ | $AP_{boundary}$↑ | mAP↑ |
|---|---|---|---|---|---|---|
| (a) | Oracle | ≤ 41 | 42.6 | 48.4 | 50.6 | 47.2 |
| | Glad (Ours) | 8 | 25.7 | 32.1 | 38.1 | 32.0 |

| | Method | Generated | $AP_{ped}$↑ | $AP_{divider}$↑ | $AP_{boundary}$↑ | mAP↑ |
|---|---|---|---|---|---|---|
| (b) | Baseline | | 42.6 | 48.4 | 50.6 | 47.2 |
| | Glad (Ours) | ✓ | 49.4 (+6.8) | 52.9 (+4.5) | 53.9 (+3.3) | 52.1 (+4.9) |

Furthermore, we report the performance of our model on online vectorized HD map construction. As shown in Tab. 4(a), for the generated validation set, our Glad achieves an mAP of 32.0, while the Oracle performance is 47.2. In Tab. 4(b), after pre-training on the generated training set, compared to the baseline, our Glad demonstrates an improvement of 4.9 in mAP.

**Potentiality for simulation.** To assess the capability of Glad as a simulator, we design an experiment to evaluate simulation stability over time. Specifically, we generate video data using reference frame, and perform 3D detection from frame 1 to frame $N$ respectively, and compare the detection performance on simulated data and real data. Fig. 4 gives the relative detection performance between simulated data and real data. The lower relative detection performance is, the simulated data collapse. Although the performance begins to decline, it quickly converges at a high percentage, such as the NDS around 85%. It demonstrates that our Glad has a good potentiality for simulation.

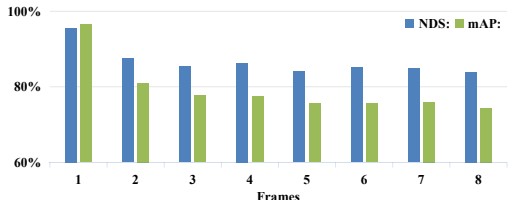

Figure 4: **The relative detection performance of simulated data compared to real data**, as the number of simulated frames increases. It tends to slightly decrease at first two frames and then becomes stable.

### 4.3 ABLATION STUDY

Table 5: **Ablation study** of our proposed method.

(a) The effectiveness of Latent Variable Propagation

| Model | FID↓ | FVD↓ | mAP↑ | NDS↑ |
|---|---|---|---|---|
| Baseline | 20.85 | - | 26.8 | 40.0 |
| +LVP | 12.57 | 207 | 28.3 | 41.3 |

(b) GPU memory consumption

| Method | 0 | 2 | 4 | 8 |
|---|---|---|---|---|
| SWS | 31GB | +14GB | +28GB | OOM |
| SDS | 31GB | +7GB | +7GB | +7GB |

(c) The impact of training video chunks

| Chunk | FID↓ | FVD↓ | mAP↑ | NDS↑ |
|---|---|---|---|---|
| 1 | 14.43 | 198 | 27.8 | 40.2 |
| 2 | 12.57 | 207 | 28.3 | 41.3 |
| 5 | 14.13 | 208 | 27.7 | 41.0 |

(d) The impact of inference video length

| Length | mAP↑ | NDS↑ | AMOTA↑ | AMOTP↓ |
|---|---|---|---|---|
| 4 | 28.1 | 41.2 | 22.1 | 1.540 |
| 8 | 28.3 | 41.3 | 22.7 | 1.526 |
| 16 | 27.7 | 40.5 | 21.9 | 1.535 |

Here we employ StreamPETR (Wang et al., 2023a) to evaluate detection and tracking performance for ablation study.

**On the LVP module.** Tab. 5(a) compares baseline and our Glad. The baseline directly generates video data from Gaussian noise, while our Glad uses latent variable propagation (LVP). Compared to the baseline, our Glad with LVP has the improvement of 1.3 and 1.5 in terms of NDS and mAP. We also report the quality of image generation, where the FID reduces from 20.85 to 12.57. These results demonstrate that the simple yet effective LVP module is vital for our Glad.

**On the SDS module.** Tab. 5(b) compares memory usage on single NVIDIA A100 GPU of streaming data sampler (SDS) in our Glad and sliding window sampler (SWS) in Panecea. Compared to SWS, our SDS does not increase memory usage with the increasing video length. The reason is that we sample video orderly at continuous iterations.

**Impact of the number of video chunks during training.** During training, we split original training video into several video chunks to balance video length and data diversity. Tab. 5(c) gives the impact of different length of video chunks. When the number of chunks is equal to 2, it has the best performance.

**Impact of the length of generated video during inference.** During inference, we can generate video data of different lengths. Tab. 5(d) gives the impact of different video lengths during inference. We observe that it has the best performance when the length is equal to 8. When the length is larger, the performance slightly drops.

**Impact of LN Layer.** We employ a LN layer to normalize the latent features as in Eq. 1. Without LN layer, we observe the training loss escalating to NAN, highlighting the importance of LN layer.

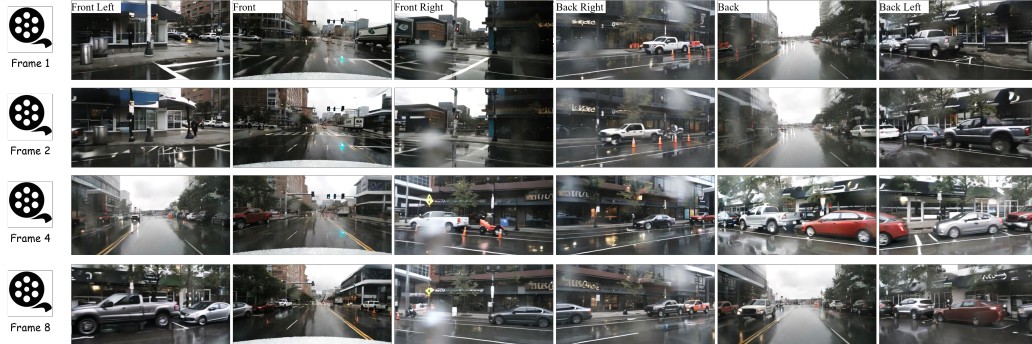

Figure 5: **Visualization of data generation examples.** We generate video clip by feeding Gaussian noise to our Glad, with BEV layout sequences starting at index 2208 of the nuScenes validation set.

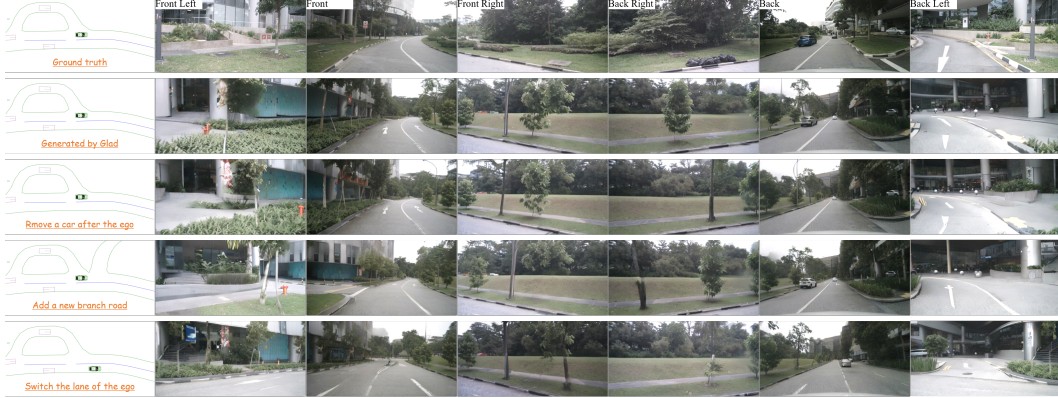

Figure 6: **Demo of editing input conditions.** The BEV's perspective visualization of input conditions is shown in the first column, where we have selected the conditions at index 74 of the nuScenes validation set.

### 4.4 QUALITATIVE ANALYSIS

Here we provide multi-view generation examples of Glad. As in Fig. 5, as a generator, our Glad generates multi-view images (corresponding to each row) with strong cross-view consistency. Moreover, Glad maintains strong temporal consistency from frame 1 to frame 8. In Fig. 6, we demonstrate the capability of generating outputs under edited input conditions using our Glad. The first row displays the ground truth as a reference. Subsequent rows show frames selected from a video based on the same reference frame, with modified conditions for each current frame. In the second row, frames are generated under original conditions. In the third row, a car following the ego is removed. In the fourth row, a new branch road is added, and in the last row, the ego vehicle switch lanes to the adjacent one. Our Glad effectively manages these conditions while maintaining consistency across frames under the same reference.

## 5 CONCLUSION

In this paper, we introduce a simple framework, named Glad, to generate video data in an online framework. Our proposed Glad extends Stable Diffusion for video generation by introducing two novel modules, including latent variable propagation and streaming data sampler. The latent variable propagation views the denoised latent features of previous frame as noise prior for image generation at current frame, leading to maintain a good temporal consistency. The streaming data sampler samples the video frame orderly at continuous iterations, enabling efficient training. We perform the experiments on the widely-used dataset nuScenes, which demonstrates the efficacy of our proposed method as a strong baseline for generation tasks.

**Limitations and future work:** We observe that our proposed method struggles to generate high-quality video data under high dynamic scenes. In these scenes, the temporal consistency of objects still needs to be improved. In future, we will explore improving high dynamic object generation.

## 6 ACKNOWLEDGE

The work was supported by National Science and Technology Major Project of China (No. 2023ZD0121300), and also supported by National Natural Science Foundation of China (No. 62271346).

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

# A APPENDIX

## A.1 PRELIMINARY: LATENT DIFFUSION MODEL

The diffusion model (Ho et al., 2020) is a sophisticated probabilistic approach for image generation. It typically encompasses two phases: the forward diffusion process in which the data is gradually injected with Gaussian noise, and the reverse process that learns to reconstruct the original data from the noise.

Specifically, in the forward diffusion process, the data $\mathbf{x}_0$ is transformed into a sequence of latent variables $\mathbf{x}_1, \ldots, \mathbf{x}_T$ following a predefined noise schedule $\beta_1, \ldots, \beta_T$:

$$q(\mathbf{x}_{1:T}|\mathbf{x}_0) = \prod_{t=1}^{T} q(\mathbf{x}_t|\mathbf{x}_{t-1}), \quad q(\mathbf{x}_t|\mathbf{x}_{t-1}) = \mathcal{N}(\mathbf{x}_t; \sqrt{1-\beta_t}\mathbf{x}_{t-1}, \beta_t \mathbf{I}). \tag{3}$$

Suppose that $p(\mathbf{x}_T) = \mathcal{N}(\mathbf{x}_T; \mathbf{0}, \mathbf{I})$. The reverse process is modeled as a Markov chain with learned transitions $\theta$, aiming to recover the original data from the noisy latent variables:

$$p_\theta(\mathbf{x}_{0:T}) = p(\mathbf{x}_T) \prod_{t=1}^{T} p_\theta(\mathbf{x}_{t-1}|\mathbf{x}_t), \quad p_\theta(\mathbf{x}_{t-1}|\mathbf{x}_t) = \mathcal{N}(\mathbf{x}_{t-1}; \boldsymbol{\mu}_\theta(\mathbf{x}_t, t), \boldsymbol{\Sigma}_\theta(\mathbf{x}_t, t)). \tag{4}$$

To maintain generation quality with limited computational resources, latent diffusion models like Stable Diffusion (Rombach et al., 2022) perform the forward and reverse processes in the latent space of pretrained VAE (Kingma & Welling, 2013). As in Fig. 2, the encoder function $\mathcal{E}$ maps the input data $\mathbf{x}_0$ to a latent space representation $\mathbf{z}_0 = \mathcal{E}(\mathbf{x}_0)$, and the decoder $\mathcal{D}$ reconstructs the data from the latent space: $\mathbf{x}_0 = \mathcal{D}(\mathbf{z}_0)$.

## A.2 MORE EXPERIMENTS AND VISUALIZATION RESULTS

Table 6: **Impact of multi-scale training on video generation.**

| Mult-scale Training | FID ↓ | FVD ↓ | mAP↑ | NDS↑ | AMOTA↑ | AMOTP↓ |
|---|---|---|---|---|---|---|
| | 12.78 | 211 | 27.9 | 41.2 | 22.4 | 1.529 |
| ✓ | 12.57 | 207 | 28.3 | 41.3 | 22.7 | 1.526 |

**Impact of multi-scale training.** Following the training configuration established in Panacea, multi-scale training is adopted as the default strategy. Specifically, the original images are randomly resized to scales ranging from 0.3 to 0.36 of their original dimensions. As demonstrated in Tab. 6, multi-scale training exhibits negligible effects on both perception and generation performance. To streamline the implementation, our proposed framework, Glad, retains multi-scale training as its default configuration.

Table 7: **Performance gains from external trajectory-based scene generation.**

| Training Data | Image Size | Backbone | Split | mAP | NDS |
|---|---|---|---|---|---|
| Real nuScenes | 512×256 | VoV-99 | val | 46.18 | 55.58 |
| Generated AV2 + Real nuScenes | 512×256 | VoV-99 | val | 46.92 | 56.64 |

**Enhancing model performance with external trajectory generation.** We explore cross-domain scenario generation by incorporating trajectories from the Argoverse 2 (AV2) dataset (Wilson et al., 2023), which contains 360-degree perception data and diverse driving scenarios. Using Far3D (Jiang et al., 2024) as the unified evaluation framework (compatible with both AV2 and nuScenes datasets), we observe 1.06 improvement in terms of NDS on the nuScenes validation set (Tab. 7). This demonstrates that trajectory-based scene generation can enhance model generalization despite domain gaps.

Table 8: **Comparison when only using generated data for training.**

| Method | nuImage | Generated | mAP↑ | NDS↑ | AMOTA↑ | AMOTP↓ |
|--------|---------|-----------|------|------|--------|--------|
| Panacea | | ✓ | 22.5 | 36.1 | - | - |
| Glad | | ✓ | 27.1 | 40.8 | 21.6 | 1.569 |
| Glad | ✓ | ✓ | 30.0 | 42.8 | 24.9 | 1.500 |

Table 9: **Per-class detection performance.** The abbreviations 'CV' and 'TC' denote Construction Vehicle and Traffic Cone, respectively.

| Method | Car | Truck | Bus | Trailer | CV | Pedestrian | Motorcycle | Bicycle | TC | Barrier | mAP |
|--------|-----|-------|-----|---------|-----|-----------|-----------|---------|-----|---------|-----|
| Oracle | 0.585 | 0.333 | 0.308 | 0.115 | 0.115 | 0.445 | 0.389 | 0.523 | 0.590 | 0.544 | 0.378 |
| Glad | 0.423 | 0.219 | 0.201 | 0.061 | 0.071 | 0.306 | 0.269 | 0.292 | 0.496 | 0.495 | 0.283 |
| Glad / Oracle (%) | 72.3 | 65.8 | 65.3 | 53.0 | 61.7 | 68.8 | 69.2 | 55.8 | 84.0 | 91.0 | 74.9 |

**Performance comparision with exclusively generated data.** In Tab. 8, we present the performance of Glad only using the generated data for training. When trained solely on synthetic data, Glad outperforms Panacea by 4.6 in terms of mAP and 4.7 in terms of NDS.

**Detail of the detection performance.** In Tab. 9, the detailed detection performance of the 10 classes can reflect the generation quality of each category. Since Panacea does not provide results broken down by category, we compare our results with the Oracle results from StreamPETR. It can be observed that common categories such as Car and Traffic cone have higher generation quality compared to rare categories like Construction Vehicles and Trailer.

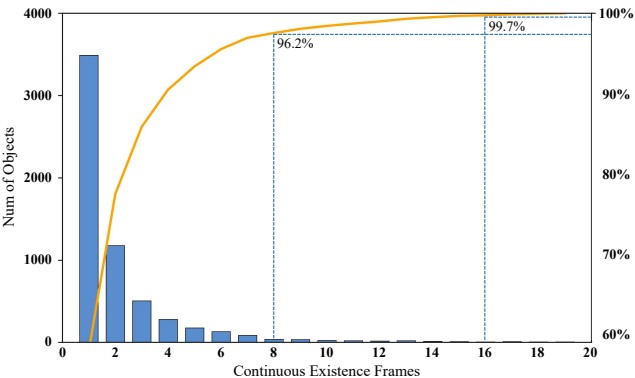

Figure 7: **Analysis of object persistence in frames.**

**Object continuous existence frames analysis.** We analyze object persistence in frames to understand why using more video frames (such as 16 frames instead of 8) does not further improve downstream task performance. As shown in Fig. 7, our analysis of the nuScenes dataset reveals that 96.2% of objects appear in fewer than 8 consecutive frames, and 99.7% appear in fewer than 16 consecutive frames.

**Ablation study on error accumulation.** As shown in Fig. 8, we evaluate error accumulation by comparing three distinct methods of modifying the noise prior. In the Tab. 10, using denoised latent features achieves the best performance in all evaluation metrics. Therefore, we utilize the denoised latent feature.

**Ablation study on temporal attention layer.** As shown in the Tab. 11, using the temporal layer for propagating temporal information yields an FID score of 17.91 and an FVD score of 183. While the FID is slightly worse than that of latent variable propagation (LVP), the FVD is marginally better. This suggests that the temporal layer more effectively models temporal propagation but has limitations in spatial modeling of single frames. Overall, LVP achieves a better balance in both spatial modeling and temporal propagation. Moreover, the approach using the temporal layer, which requires modeling multiple frames simultaneously, is considerably more expensive in terms of memory consumption and training time.

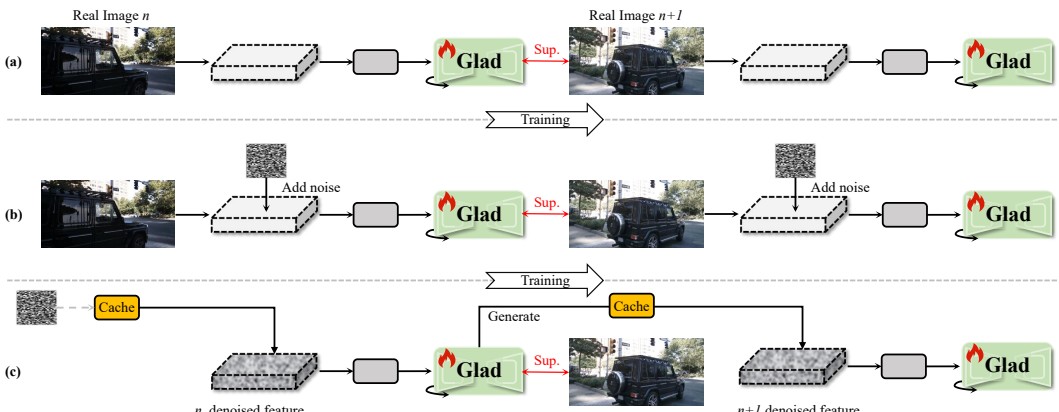

Figure 8: **Different noise prior.** Here, "Sup." refers to the training supervision for Glad, with LayerNorm omitted for clarity. In (a), we directly use the latent feature of real image n as the noise prior. In (b), we add Gaussian noise to the latent feature, while our Glad in (c) directly denoises and caches latent features.

Table 10: **The influence for error accumulation under different noise prior.** Three strategies correspond to Fig. 8, the strategy (a) is the latent freatures of privious frame, the strategy (b) is the noise added latent freatures of privious frame, and the strategy (c) is newly generated by our Glad.

| Strategy | FID ↓ | FVD ↓ | mAP↑ | NDS↑ |
|----------|-------|-------|------|------|
| (a) | 49.02 | 607 | 15.5 | 30.9 |
| (b) | 32.43 | 362 | 19.5 | 35.5 |
| (c) | **12.57** | **207** | **28.3** | **41.3** |

Table 11: **Performance with temporal attention layer.**

| Temporal modeling | FID ↓ | FVD ↓ | Memory↓ | Training Time↓ |
|-------------------|-------|-------|---------|----------------|
| Latent Variable Propagation (LVP) | 12.57 | 207 | 38GB | 1 Days |
| Temporal Layer (TL) | 17.91 | 183 | 66GB | 5 Days |

**Visualization results on corner case scenes.** To generate corner cases, we utilize the CODA (Li et al., 2022a) dataset, which is specifically designed for real-world road corner cases. CODA encompasses three major autonomous driving datasets, including 134 scenes from nuScenes. We focus on this subset of 134 nuScenes scenes. After filtering to identify which scenes belong to the validation set, we retain 30 valid scenes. These 30 scenes serve as our corner case examples, and we present selected generation results in Fig. 9.

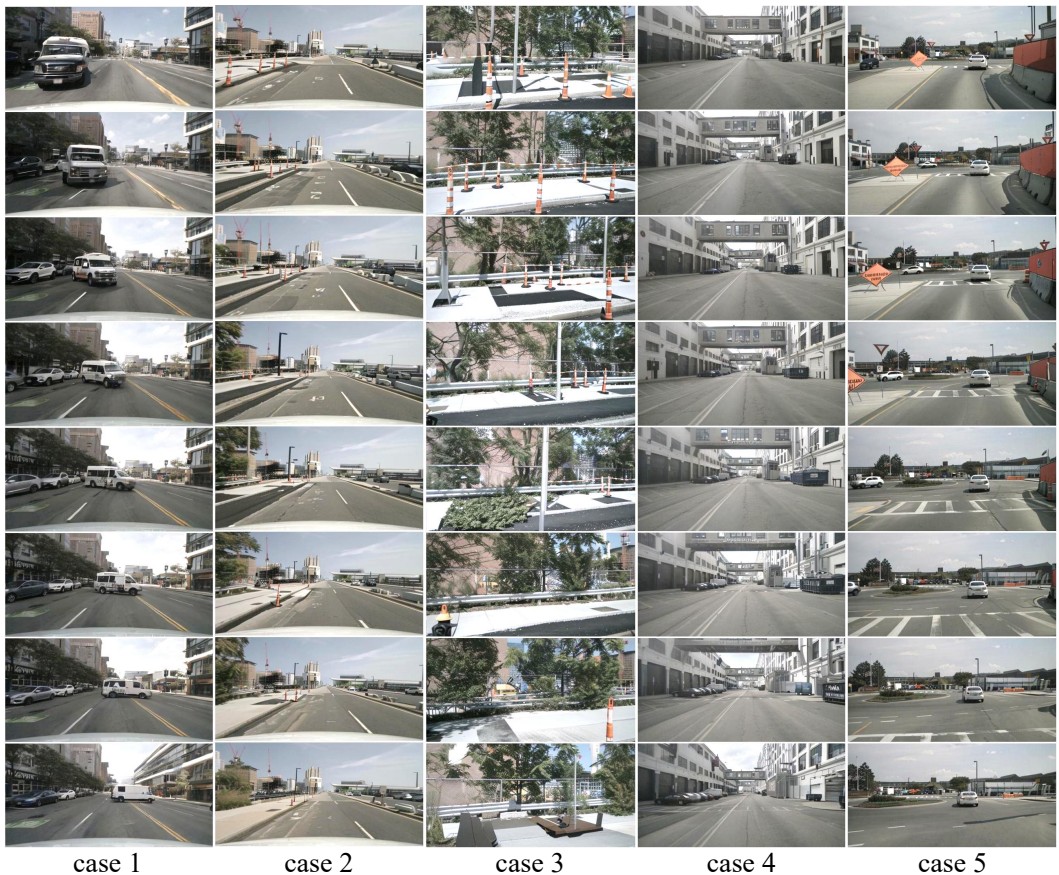

case 1  case 2  case 3  case 4  case 5

Figure 9: **Corner cases generation.** From the nuScenes validation dataset, we showcase five corner cases that demonstrate our generation capabilities: a car making a sudden turn in the opposite lane (case 1), traffic cones along the roadside (cases 2 and 3), unusual roadside obstacles (case 4), and unconventional road signs (case 5).

