# OpenReview forum: "Glad: A Streaming Scene Generator for Autonomous Driving"
_ICLR.cc/2025/Conference — ICLR 2025 Poster_

### Official Review · Reviewer_MnsB · 2024-10-31

**Soundness:** 3
**Presentation:** 3
**Contribution:** 2
**Rating:** 6
**Confidence:** 4

**Summary:**

The paper introduces Glad, a novel framework designed to generate and simulate video data in an online setting, addressing the limitations of existing methods in handling unseen scenarios and short video lengths. Glad ensures temporal consistency in the generated videos by incorporating a hidden variable propagation module, which uses the denoised hidden variable features from the previous frame as a noise prior for the current frame. The framework includes a stream data sampler that sequentially and iteratively samples frames from video clips, enabling efficient training and enhancing the model's ability to generate specific scene videos.

**Strengths:**

1. Glad stands out by enabling online video generation, over traditional offline methods that are limited to fixed-length video generation.
2. By treating the denoised hidden variable features from the previous frame as a noise prior for the current frame, Glad maintains a high level of temporal coherence, which is essential for realistic and seamless video generation.
3. The Stream Data Sampler (SDS) is designed to improve training efficiency by sequentially sampling frames from video clips across multiple iterations. This method not only ensures consistency between training and inference but also optimizes the training process by caching the generated hidden features for use as noise priors in subsequent iterations.
4. Glad integrates both data generation and simulation capabilities within a unified framework, offering a comprehensive solution for autonomous driving applications. Whether generating new scene videos from noise or specific scene videos given a reference frame, Glad demonstrates versatility and effectiveness.

**Weaknesses:**

1. The paper claims significant advancements in online generation of controllable videos for driving simulation, yet the supporting evidence is lacking. The experiments presented in Fig. 5&6 do not convincingly demonstrate the quality of video generation under novel trajectories from a scientific perspective. It is recommended that the authors provide additional qualitative analysis of Glad's performance under different novel trajectories (e.g.,  sharp turns, lane changes, or varying traffic densities) starting from the same initial frame to substantiate their claims.
2. The paper does not clearly articulate the fundamental benefits of using past frames as priors for current frame generation compared to methods that incorporate temporal transformer layers, such as Panacea and DrivingDiffusion. An ablation study to validate the effectiveness of this approach is warranted, especially given that Tab. 1 shows Panacea achieving better video consistency (FVD) without the use of an initial frame. The authors can compare their approach with temporal transformer layers with Panacea/DrivingDiffusion, focusing on FID, FVD and computational efficiency.
3. The authors do not explore or discuss the potential performance gains that could be achieved by training models on data generated with new trajectories. This is a significant oversight that limits the understanding of Glad's full potential. The authors may generate an additional batch of data using novel trajectories rather than original trajectories, use it as a training data for Tab. 3, and then see if this dataset produces a gain in the model.
4. The training data used in the stage one is not detailed in the paper. At L285, the authors fail to specify whether the data comes from public datasets, web sources, self-collected data, or a combination thereof. This lack of transparency raises concerns about privacy issues, the necessity of data annotation, and the distribution of the data. The authors can provide a detailed breakdown of data sources, including preprocessing steps, annotation methods, and measures taken to address privacy concerns.

**Questions:**

My concern revolves around what I perceive as an overclaim by the authors regarding the simulation capabilities of driving video generation. How effectively the authors address this critical point will significantly influence my score on their work.

---

> ### Author Response · Authors · 2024-11-23
> **Official Comment by Authors**
>
> We thank reviewer MnsB for the valuable time and constructive feedback.
> ***
> **W1: On the lack of supporting evidence.**
>
> **A1:** Thank you for your suggestions regarding qualitative analysis. We have provided demonstrations for various scenarios: lane changes can be seen in the last column of **Fig. 6 in the manuscript** (achieved by manually changing ego position), sharp turns are shown in **Fig. 9 in the Appendix** (focusing on corner cases), and examples of varying traffic densities are available in the supplementary material.
> ***
> **W2: About the ablation with temporal transformer layer.**
>
> **A2:** Thank you for your suggestion. Following the implementation of Panacea, we have incorporated a temporal layer into our framework for an ablation study. As shown in the table below, using the temporal layer for propagating temporal information yielded an FID score of 17.91 and an FVD score of 183. While the FID is slightly worse than that of latent variable propagation (LVP), the FVD is marginally better. This suggests that the temporal layer more effectively models temporal propagation but has limitations in spatial modeling of single frames. **Overall, LVP achieves a better balance in both spatial modeling and temporal propagation.** Moreover, the approach using the temporal layer, which requires modeling multiple frames simultaneously, is considerably more expensive in terms of memory consumption and training time. Therefore, the LVP method is more efficient. We have updated these ablation results in the Appendix.
>  Temporal modeling | FID ↓ | FVD ↓ | Memory ↓ | Training Time ↓ |
> |:----------------:|:-----:|:-----:|:--------:|:---------------:|
> | Latent Variable Propagation (LVP) | 12.57 | 207 | 38GB | 1 Days |
> | Temporal Layer (TL) | 17.91 | 188 | 66GB | 5 Days |
>
> ***
> **W3: About the performance gain by generating new trajectories.**
>
> **A3:** We have taken your suggestion seriously. However, there are some problems when we consider conducting the experiment. First, there is no open-source toolkit available to edit or generate new trajectories for the nuScenes dataset. Generating additional data require the manual annotation of a large number of trajectories, which would be extremely costly. Second, simply generating new trajectories may pollute the existing data, as the new trajectories may not follow the physical world—e.g., the trajectory might go beyond the road or make impossible turns. A simulator environment for trajectory generation that complies with physics is urgently needed for current video generation models for autonomous driving. There are some relevant works, such as HouseLLM[1], focused on the generation of indoor layouts. We eagerly anticipate the extension of similar work to the domain of autonomous driving in future research.
> ***
> **W4: About the training data.**
>
> **A4:** We apologize for the lack of detail about the training data, which we have now included in the manuscript. Our two-stage training process utilizes the same data and processing rules as Panacea. We only utilized key frames (2HZ) that have detailed 3D bounding box annotations. In the first stage, we conduct resizing and cropping operations solely on single-frame images. Considering the native resolution of $1600\times900$, we randomly resize the images to a proportion of 0.3 to 0.36, and then crop a $512\times256$ region. In the second stage, we further organized the nuScenes dataset into a video stream format to facilitate the fine-tuning of streaming video generation.
> ***
> **Q1: About the simulation capabilities.**
>
> **A5:** Thank you for your concern. To demonstrate the simulation capability, we use the end-to-end autonomous driving model VAD[2] as an example, retraining it at $512\times 256$ resolution. We perform inference using this retrained model on the validation set generated by Glad and include the recorded process in various scenes as **a video in our supplementary material.** It's worth noting that simulators based on video generation models are still in their early stages. While generative models excel at creating new scenes compared to fully NeRF-based simulator methods, NeRF-based models offer superior reconstruction of existing scenes with lower computational costs. Recent work has explored combining these approaches, using generative models to create unseen or low-quality scenes that NeRF-based models can then reconstruct more effectively. We believe generative models show significant promise for developing non-rule-based simulators.
> ***
> We appreciate your thoughtful review and we hope to address your concerns. Please let us know if you'd like any further discussion.
> ***
>
> [1]Zong, Ziyang, Zhaohuan Zhan, and Guang Tan. "HouseLLM: LLM-Assisted Two-Phase Text-to-Floorplan Generation." arXiv preprint arXiv:2411.12279 (2024).
>
> [2]Jiang, Bo, et al. "Vad: Vectorized scene representation for efficient autonomous driving." Proceedings of the IEEE/CVF International Conference on Computer Vision. 2023.

---

> > ### Comment · Reviewer_MnsB · 2024-11-25
> >
> > Thank you for your rebuttal, it addresses most of my concerns. I regret the limitation within W3 pertaining to the validation of performance enhancements through the creation of new scenarios. I agree that there are no tools available for a reasonable batch generation of new trajectories. However, this does affect the practicality of the paper, as scene editing remains limited to the demos. Perhaps there are alternative solutions from simulators that could enable batch generation of new scenarios, such as Scenarionet. Have you contemplated the integration of such methodologies? Please correct me if I am wrong.

---

> ### Author Response · Authors · 2024-11-28
> **Follow-up Official Comments by Authors**
>
> Dear Reviewer,
>
> Thank you for your valuable suggestions, especially for introducing ScenarioNet to us. We have taken a close look at ScenarioNet, which collects trajectories from several public datasets and reconstructs them within a simulator. This approach has greatly inspired us, prompting us to attempt the generation of augmented data using trajectories from other datasets as new trajectories. Specifically:
>
> We selected trajectories from the Argoverse 2[1] (AV2) dataset to serve as our new trajectories due to its capability to encompass a full 360-degree view. Subsequently, we employed Glad to infer over the trajectories of the AV2 dataset to obtain new generated scenes, which were then integrated with the nuScenes dataset to enhance performance. We chose Far3D[2] as our perception evaluation method, as it offers training support for both the AV2 and nuScenes datasets. The experimental results are as follows:
>
> | Training data | Image Size | Backbone | Split | mAP | NDS |
> | --- | --- | --- | --- | --- | --- |
> | real nuScenes | 512x256 | VoV-99 | val | 0.4618 | 0.5558 |
> | generated AV2 + real nuScenes | 512x256 | VoV-99 | val | 0.4692 | 0.5664 |
>
> As shown in the table, we observed that utilizing new trajectories for data generation leads to performance improvements (about 1% NDS). Notably, Far3D employs a stronger backbone, VoV-99, which has been pre-trained on the DD3D 15M dataset. Moreover, the camera poses and scene distributions in AV2 differ from those in nuScenes. In the future, if specialized tools for trajectory generation become available, creating in-domain trajectories could potentially yield even better results.
>
> We hope to address your concerns. Please let us know if you'd like any further discussion.
>
> Best regards.
> ***
> [1] Wilson, Benjamin, et al. "Argoverse 2: Next generation datasets for self-driving perception and forecasting." arXiv preprint arXiv:2301.00493 (2023).
>
> [2] Jiang, Xiaohui, et al. "Far3d: Expanding the horizon for surround-view 3d object detection." Proceedings of the AAAI Conference on Artificial Intelligence. Vol. 38. No. 3. 2024.

---

> > ### Comment · Reviewer_MnsB · 2024-12-02
> >
> > Thanks for your rebuttal. It solvs most of my concerns and I will bump up the score. If this paper ends up being accepted please make sure to include these expeeriments in the paper.

---

> > > ### Author Response · Authors · 2024-12-02
> > > **Follow-up Official Comments by Authors**
> > >
> > > Dear Reviewer,
> > >
> > > Thank you for your decision. We are pleased that our response has addressed your concerns, and we will include these experiments in our revised paper.
> > >
> > > Best regards.

---

### Official Review · Reviewer_XvyP · 2024-11-02

**Soundness:** 2
**Presentation:** 2
**Contribution:** 2
**Rating:** 5
**Confidence:** 5

**Summary:**

This paper generates multi-view video data in a frame-by-frame style for autonomous driving.

To be honest, I reviewed this paper at NIPS, and seeing it again, I am pleased that the authors have taken into account previous reviewers' feedback, such as modifying the wording and adding discussions about previous methods. However, since there were no changes made to the methodology, considering the limited novelty and suboptimal results, I will maintain a negative score.

**Strengths:**

1. This paper proposed Glad, which generates multi-view video data in a frame-by-frame style and can generate videos of arbitrary lengths theoretically.

2. The paper proposes two training tricks, i.e., Latent Variable Propagation and Streaming Data Sampler, to achieve frame-by-frame generation.

3. The tables and figures are well-presented.

**Weaknesses:**

1. Latent Variable Propagation and Streaming Data Sampler appear to be simple training tricks. Many previous U-Net-based works with temporal modules (before Sora and SVD), such as ADriver-I, MagicDrive, and Drive-WM, also employ frame-by-frame video generation strategy. Many works in the video generation area also use cross-frame attention of each frame on the first/former frame to preserve the context, appearance, and identity consistency of the video: Text2Video-Zero: Text-to-Image Diffusion Models are Zero-Shot Video Generators; ControlVideo: Training-free Controllable Text-to-Video Generation; Video ControlNet: Towards Temporally Consistent Synthetic-to-Real Video Translation Using Conditional Image Diffusion Models.

2. The experimental results should provide a comparison between the proposed approach and previous methods, e.g., DriveDreamer-2, considering that DriveDreamer-2 has achieved significantly better results with an FVD of 55.7 compared to the author's Glad approach with 206.

**Questions:**

See Weaknesses.

---

> ### Author Response · Authors · 2024-11-23
> **Official Comment by Authors**
>
> Thank you for taking the time to review our paper again and providing your valuable feedback. We appreciate your insights and the opportunity to address your concerns. In response to your comments, we would like to offer the following clarifications:
> ***
> **Summary: On the limited novelty and suboptimal results.**
>
> **A1:** Thank you for serving as this paper's reviewer again. Regarding the limited novelty, even though our Glad is simple, it proposes a new paradigm. Compared to current methods that depend on temporal attention layers, Glad shows that the importance of the noise prior for video consistency has been ignored. By the way, our latent variable propagation is easy to follow. We think the contribution of our Glad is to tell the research community to pay more attention to noise priors. As for the suboptimal results, the mainstream benchmark of current video generation is FVD. The FVD lacks a uniform platform or requirements; in our practical experience, there are many tricks to reduce the FVD score, such as adjusting the resolution of the video, the length of the video, and the FPS of videos. Finally, we want to highlight that Glad is an initial attempt to steer research towards paying more attention to the noise prior.
> ***
> **W1: About the simple training tricks for our proposed modules.**
>
> **A2:** Thank you for your feedback.
> Although the streaming data sampler is concise and intuitive, it is indeed a new training paradigm. It is designed for our frame-by-frame video generation. For ADriver-I, MagicDrive, and Drive-WM, they are not truly frame-by-frame. For example, ADriver-I inputs 4 frames to generate the future 4 frames, MagicDrive generates 7-frame clips at once, and Drive-WM also adopts 3D-UNet for multi-frame generation at once. Not only that, the sampler is also efficient for alleviating error accumulation, and as shown in Tab.5(b), this sampler can reduce the GPU memory requirement. For the previous works that also adopt first/former frame to preserve the context, Text2Video-Zero just backpropagates the Gaussian noise to maintain the inner-frame consistency, which is not for preserving the context, and it also integrates Cross-Frame attention. ControlVideo appears to just use original Gaussian noise. Video ControlNet explores optimizing the initial Gaussian noise to reduce drastic changes in texture, which is not designed for frame-by-frame video generation.
> ***
> **W2: On the comparison with previous methods.**
>
> **A3:** We have expanded comparisons with both earlier and contemporary works, such as Vista and DriveWM. For DriveDreamer-2, the notably low FVD performance was also of interest to us. **Delving into the experimental details, we found that its training and testing settings differ from the mainstream methods.** In contrast, our approach has maintained the same training and testing settings as Panacea. The specific differences are as follows:
>
> DriveDreamer-2 has annotated the sweep data from the nuScenes dataset to create a 12Hz training and testing dataset, while other methods like Panacea and MagicDrive have only used 2Hz keyframe data. This means DriveDreamer-2 has utilized 6 $\times$ more training data than these methods.
> For FVD testing, a sequence of 16 frames is typically required. Since DriveDreamer-2 tests at 12Hz, it only needs to generate a 1.33 seconds video clip. In comparison, Glad and Panacea, must maintain a video length of 8 seconds. Moreover, DriveDreamer-2 uses real-scene images as the initial frame. With a video length of just 1.33 seconds, there is minimal scene movement, making it easier to maintain consistency with the initial frame. These factors may contribute to the low FVD score of DriveDreamer-2.
> ***
> We appreciate your thoughtful review and we hope to address your concerns. Please let us know if you'd like any further discussion.
> ***

---

> > ### Comment · Reviewer_XvyP · 2024-11-25
> >
> > For A1:
> > In the field of video generation, the Cross-Frame Attention module in a series of papers (such as Panacea, Drive-WM, and Drive-Dreamer) fundamentally leverages latent features from the previous frame to enhance the spatiotemporal consistency of the current frame, much like your Latent Variable Propagation module. It would be beneficial for the authors to provide a more detailed explanation of how Glad utilizes previous latent features compared to traditional cross frame attention mechanisms and other related works, to further elucidate Glad's contributions.
> >
> > For A2:
> > Utilizing a Streaming Data Sampler is a common practice in frame-by-frame video generation methods. Papers like Panacea, Drive-Dreamer, and Drive-WM are all based on UNet for generating videos frame by frame, hence naturally necessitating the use of streaming, which essentially involves sequential sampling within clips. If they are not utilizing streaming training, it would be helpful for the authors to clarify the points of differentiation.
> >
> > For A3:
> > Aligning the settings of different approaches for a dedicated comparison would be beneficial.
> >
> > Summary:
> >
> > I still believe that this paper simply combines two common tricks without demonstrating satisfactory results.

---

> > > ### Author Response · Authors · 2024-11-27
> > > **Follow-up Official Comments by Authors**
> > >
> > > Dear Reviewer,
> > >
> > > Thank you for your additional feedback on our paper. We have carefully reviewed your comments and provide the following responses:
> > > ***
> > > **W1: Differences between Cross-Frame Attention and Latent Variable Propagation.**
> > >
> > > **A1:** We are glad to explain the key difference between Cross-Frame Attention (CFA) and Latent Variable Propagation (LVP):
> > > -	CFA primarily focuses on **temporal modeling through feature propagation**, predominantly utilized in the reverse process of diffusion. It employs an attention mechanism to propagate information from the initial frame across other frames within the video clip, thereby ensuring inter-frame continuity.
> > > -	LVP, on the other hand, approaches **temporal propagation from the perspective of noise distribution. Its core assumption is that the latent features of frame T-1 can serve as the noisied features for frame T.** By denoising the features of frame T-1, we can restore the features of frame T, which is reasonable. During training, Glad generate noise based on the latent distribution of frame T - 1 to replace the Gaussian noise in the forward diffusion process and gradually perform denoising in the reverse process.
> > >
> > > When considering only the aspect of temporal modeling, neither CFA nor LVP module is inherently complex. However, the modeling approach that incorporates Cross-Frame Attention often requires a large video clip window to ensure that the attention network can learn temporal consistency. This requirement significantly increases the training costs and memory consumption. In contrast, LVP is considerably more cost-effective, offering a lower computational overhead.
> > >
> > > ***
> > >
> > > **W2: About the Streaming Data Sampler**
> > >
> > > **A2:** Since we cannot insert images directly here, we have sketched the training and testing pipelines of both paradigms below. As shown in the table below, current methods like Panacea and Adriver-I sample multiple consecutive frames ($n=8$ or $16$) during each training iteration, then randomly sample another set of consecutive frames for the next iteration. In contrast, during the training stage, our Glad only needs to process one frame at a time using the Streaming Data Sampler, making it more efficient.
> > >
> > > |Train Time | Adriver-I / Panacea                           | Glad                           |
> > > |--------------------|------|----------|
> > > | iter 1        | T1 T2 T3 T4 ----  add noise, denoising ----> T1 T2 T3 T4  |T2 ----  T1 noise prior, denoising ----> T2  |
> > > | iter 2        | T2 T3 T4 T5 ----  add noise, denoising ----> T2 T3 T4 T5   |T3 ----  T2 noise prior, denoising ----> T3  |
> > > | iter 3        | T3 T4 T5 T6 ----  add noise, denoising ----> T3 T4 T5 T6   |T4 ----  T3 noise prior, denoising ----> T4  |
> > >
> > > |Test Time | Adriver-I / Panacea           | Glad      |
> > > |--------------------|------|----------|
> > > | infer 1        | T1, Noise, Noise, Noise ----> T1 T2 T3 T4   | T1 ----> T2  |
> > > | infer 2        | T2, Noise, Noise, Noise ----> T2 T3 T4 T5   | T2 ----> T3  |
> > > | infer 3        | T3, Noise, Noise, Noise ----> T3 T4 T5 T6   | T3 ----> T4  |
> > > ***
> > > **W3: Aligning the settings would be beneficial**
> > >
> > > **A3:** We agree with your comments. Since the results for DriveWM and MagicDrive have already been updated in the manuscript in previous response, the primary remaining comparison is with DriveDreamer-2.
> > >
> > > The nuScenes dataset only provides annotations for keyframes (at 2Hz). Consequently, BEVControl, MagicDrive, Panacea, and DriveWM all train and test only on these keyframes. DriveDreamer-2, on the other hand, has its own annotations created for the sweep data (12 Hz), which is not consistent with the settings of these other methods. Since DriveDreamer-2 is not open-sourced, we are unable to retrain and test it on keyframes. We are willing to conduct this experiment and update our comparative experiments once it becomes publicly available in the future.
> > > ***
> > > We appreciate your thoughtful review and we hope to address your concerns. Please let us know if you'd like any further discussion.
> > >
> > > Best regards.

---

> > > > ### Comment · Reviewer_XvyP · 2024-12-02
> > > >
> > > > Sorry for the late reply.
> > > >
> > > >
> > > > **W1: About Latent Variable Propagation**
> > > >
> > > > The LVP module generates noise based on the latent distribution of frame T-1 to replace Gaussian noise during the forward diffusion process. This methodology **resonates with modules utilized in previous papers** such as "PYoCo" and "VideoFusion," where noise priors have been extensively investigated.
> > > >
> > > > Furthermore, **is there a theoretical basis for this substitution**, or is it primarily a practical engineering strategy?
> > > >
> > > > References:
> > > >
> > > > - "Preserve Your Own Correlation: A Noise Prior for Video Diffusion Models"
> > > > - "VideoFusion: Decomposed Diffusion Models for High-Quality Video Generation"
> > > >
> > > > **W2: Concerning the Streaming Data Sampler**
> > > >
> > > > | Test Time | Adriver-I / Panacea                       | Glad        |
> > > > | :-------- | :---------------------------------------- | :---------- |
> > > > | infer 1   | T1, Noise, Noise, Noise ----> T1 T2 T3 T4 | T1 ----> T2 |
> > > > | infer 2   | T2, Noise, Noise, Noise ----> T2 T3 T4 T5 | T2 ----> T3 |
> > > > | infer 3   | T3, Noise, Noise, Noise ----> T3 T4 T5 T6 | T3 ----> T4 |
> > > >
> > > > The pipeline you provided, as shown in the table above, contrasts with the approach described in the Panacea GitHub, where *--use_last_frame=true* indicates the use of the last frame as a conditional image. The revised pipeline, as depicted in the subsequent table, appears considerably **more efficient than the method proposed**.
> > > >
> > > > | Test Time | Adriver-I / Panacea (You Provide)              | Glad            |
> > > > | :-------- | :--------------------------------------------- | :-------------- |
> > > > | infer 1   | T1, Noise, Noise, Noise ----> T1 T2 T3 T4      | T1 ----> T2     |
> > > > | infer 2   | T4, Noise, Noise, Noise ----> T4 T5 T6 T7      | T2 ----> T3     |
> > > > | infer 3   | T7, Noise, Noise, Noise ----> T7 T8 T9 **T10** | T3 ----> **T4** |
> > > >
> > > > **W3: Aligning the settings would be beneficial**
> > > >
> > > > I understand that FVD may not always be a convincing metric, but how can the superiority of the method be demonstrated? Even from the carefully provided demos by the authors, it's evident that **the consistency is very poor**.

---

> > > > > ### Author Response · Authors · 2024-12-03
> > > > > **Follow-up II Official Comments by Authors**
> > > > >
> > > > > Dear Reviewer,
> > > > >
> > > > > Thank you for taking the time to review our latest response. We are willing to engage in further discussion with you.
> > > > > ***
> > > > > **W1:  About Latent Variable Propagation**
> > > > >
> > > > > **A1:** While "noise priors" have been used in works like "PYoCo" and "VideoFusion", in our framework they function as an intermediate representation within the LVP module. The essential contribution of LVP lies in the exploration of frame-by-frame propagation leveraging noise priors. Moreover, there is a significant difference in how noise priors are applied in these studies. "PYoCo" and "VideoFusion" emphasize that noise distributions across multiple frames should be correlated rather than independent, focusing on noise sampling and clip-based video generation.  In contrast, Glad assume that the latent features of frame T-1 can serve as the noisy features for frame T, employing an autoregressive-like form of propagation.
> > > > >
> > > > > For the theoretical basis, Glad is based on diffusion models and employs an autoregressive-like approach for frame-by-frame generation. We will include more rigorous theoretical substantiation in the Appendix section of our revised manuscript.
> > > > > ***
> > > > > **W2: Concerning the Streaming Data Sampler**
> > > > >
> > > > > **A2:** Thank you for your meticulous attention to these details. Our last response was primarily focused on explaining the frame-by-frame approach with streaming data sampler. In fact, as you noted in the table above, Panacea and Adriver-I do not utilize a frame-by-frame generation method. Frame-by-frame generation is more significant for world models and end-to-end autonomous driving scenarios. In end-to-end autonomous driving, decisions may dynamically change within each frame. Although Panacea can infer multiple frames at once, only the next frame is likely to be utilized, as depicted in the table we provided in our last response. Under such settings, the efficiency of Panacea's generation is considerably diminished compared to the flexibility offered by Glad.
> > > > > ***
> > > > > **W3: Aligning the settings would be beneficial**
> > > > >
> > > > > **A3:** We acknowledge that our FVD results did not achieve state-of-the-art (SoTA) performance, which is why we did not make such claims in the main text. While FVD is one of several metrics for assessing temporal performance, our method demonstrated superior results in downstream temporal perception tasks, particularly on the generated validation set.
> > > > >
> > > > > Regarding our generated results' visualization, we observed that creating continuous 40-frame video sequences results in some color fluctuations in objects—a limitation also present in previous methods. We apologize that these results did not meet your expectations.
> > > > > ***
> > > > >
> > > > > We hope to address your concerns. Please let us know if you'd like any further discussion.
> > > > >
> > > > > Best regards.

---

> > ### Comment · Reviewer_wGv7 · 2024-11-26
> >
> > Sorry to jump in. I am aware that many video generation method does not have a reasonable toolkit to benchmark the generation quality, but if your guess is they use more data, how about use the same amount of data to train GLAD and see the results?

---

> > > ### Author Response · Authors · 2024-11-27
> > > **Follow-up Official Comments by Authors**
> > >
> > > Dear Reviewer,
> > >
> > > Thank you for your feedback. We have carefully reviewed your comments and provide the following response.
> > >
> > > The nuScenes dataset provides keyframe annotations at 2Hz. Most existing approaches, including BEVControl, MagicDrive, Panacea, and DriveWM, have aligned their training and evaluation protocols with these keyframes. While DriveDreamer-2 (see Section 4.1 of their paper) has created its own annotations for sweep data at 12Hz, making its experimental settings different from other methods, this creates a different experimental setting that makes direct comparisons with other methods challenging. We are willing to retrain DriveDreamer-2 on keyframes for a fair comparison, but unfortunately, it has not been open-sourced yet. However, achieving higher performance through additional annotations and trick settings should not be encouraged by the community.
> > >
> > > We hope to address your concerns. Please let us know if you'd like any further discussion.
> > >
> > > Best regards.

---

> > > > ### Comment · Reviewer_8mSK · 2024-11-30
> > > >
> > > > Sorry for the jump in, but I tend to agree with authors about the low FVD of DriveDreamer2 come from the 12Hz video generation settings.
> > > > As a matter of fact, FVD just measures the clip-level mean feature variance between of the original videos and the generated videos.
> > > > A shorter generation time span will has an exceptional advantage over all other methods and only DriveDreamer2 adopts this weird setting.

---

> > > > ### Comment · Reviewer_wGv7 · 2024-12-01
> > > >
> > > > Sorry for the late reply. My suggestion is to train your method with their setting not on the other way round. Using purely additional trick should not be encouraged or in other words should be pointed out and criticize, one should also use facts rather than guesses to validate the serious claim.
> > > >
> > > > So my question still remains unaddressed, how does your method perform under their setting, using 12 Hz? If the method is better, you should see also an improvement right?
> > > >
> > > > All in all, if you would like to argue the drivedreamer-2 is cheating, this is valuable and all practitioners will welcome this. But we need concrete analysis and proof.

---

> > > > > ### Author Response · Authors · 2024-12-03
> > > > > **Follow-up II Official Comments by Authors**
> > > > >
> > > > > Dear Reviewer,
> > > > >
> > > > > Thank you for your reply. We appreciate the opportunity to clarify a technical detail regarding the dataset again. The official nuScenes dataset provides annotations at 2Hz, while DriveDreamer-2 utilizes private 12Hz annotations. Given that DriveDreamer-2's implementation is not publicly available and uses a different annotation protocol, we are unable to make direct comparisons at 12Hz. Although DriveDreamer-2's FVD results are impressive, we maintain that adhering to the standard 2Hz protocol better serves the research community. The responsibility for comparing 12Hz and 2Hz performance lies with DriveDreamer-2, not our current work. It is worth noting that even after half a year, there is still no method that surpasses DriveDreamer-2 in terms of FVD performance.
> > > > >
> > > > > We hope to address your concerns. Please let us know if you'd like any further discussion.
> > > > >
> > > > > Best regards.

---

### Official Review · Reviewer_wGv7 · 2024-11-04

**Soundness:** 2
**Presentation:** 2
**Contribution:** 2
**Rating:** 3
**Confidence:** 4

**Summary:**

This paper proposes “Glad”, to generate visual data from scenarios of autonomous driving in a frame-by-frame manner, where most of the previous works generate the entire sequence at a certain amount of time. It is done by injecting the previous frame noise into later frames. In addition, it designs a continuous sampling the original images into a sequence. Experiments are conducted on nuScenes dataset for video generation, and boosting 3D detection.

**Strengths:**

This paper is overall easy to read.

Use a previous frame noise to initiate the following frame noise seems to be a good way to maintain the consistency

The streaming data sampler seems to be a good engineering approach to cache the previous timeframe’s noise and inject into the second iteration instead of recomputing.

**Weaknesses:**

I have several concerns regarding this paper’s motivation, technical novelty, experiments and presentations.

## Motivation concerns

In my humble opinion, this paper has makes a seemingly interesting claim for generating scenes in a frame-by-frame manner but unclear reason why we should do this.

For example, the authors claim in the abstract that “these approaches suffers from unseen scenes or restricted video length”. However, as I understand correctly, the recent diffusion based method can all support such sampling strategy to realize infinite sequence generation. For example, Drive-WM (CVPR2024) and Panacea (CVPR2024) also use the generated frame as the subsequent condition for long video generation. As for unseen scenes, as a layout-dependent generation framework, people in general can pass any unseen layout to generate such unseen scenes. A most recent one [A], although this one is still an Arxiv paper, shows the capability of generating such unseen scenes to boost end-to-end algorithms. Though this cannot be used to directly penalize this work is wrong, it shows that previous approaches can indeed generate unseen scenes, which contradicts to the paper’s core motivation.

[A] Ma et al. Unleashing Generalization of End-to-End Autonomous Driving with Controllable Long Video Generation, arxiv


The second motivation is depicted in Figure 1 (b-c). This paper claims that previous papers focused on offline generate scenes. If I understand correctly, this means previous method takes a pre-defined trajectory, while this paper can take a dynamic trajectory. However, is this merely an engineering approach? After all, I can also passed a more flexible trajectory to the previous generation method.

In addition, even though this motivation is true when you try to encode this “Glad” method into some rendering engine to build a close loop simulator, I did not see any of such validation in the latter experiment section. So I am not convinced this is a valid motivation.

All in all, this paper’s motivations need some further clarification.


## Technical novelty concerns

Is this using a single frame as condition to generate next frame already done in DriveWM and Panacea, by setting the sequence length to 1 frame and keep generating? Please clearly discuss what are the differences between their’s approach and this one.

Streaming data sampler is an engineering effort without much technical novelty. It is just a simple caching mechanism during the python programming.

## Empirical validation concerns


As aforementioned, the motivation of injecting dynamic scenes is not well justified during the experiment sections. The only downstream applications other than video quality is 3D object detection. Please at least use end-to-end autonomous driving or 3D object tracking tasks to showcase if the motivation is validated.

In addition, there is no validation on long sequence video quality.

Also, does the word “online” meaning that on-the-fly rendering? Can you show the complexity and time benchmarking?

## Presentation concerns

This paper made simple mistakes in presentation. For example, it should use ‘citep’ when mentioning a method in a third perspective rather than use ‘cite’ like CVPR template.

**Questions:**

as above weaknesses

---

> ### Author Response · Authors · 2024-11-23
> **Official Comment by Authors**
>
> We thank reviewer wGv7 for the valuable time and constructive feedback.
> ***
> **W1: On the motivation concerns.**
>
> **A1:** We are sorry about the lack of clarity in our motivations.
> Regarding the restricted video length claimed in our paper, we do not intend to question the ability of current methods for long video generation. Instead, we focus on the efficiency and potential to effectively conduct long video generation between two different paradigms. More specifically, Drive-WM generates long videos by first generating frames 1 to N, and then frames 2 to N+1. This means generating N frames at once but only keeping one frame, which is inefficient. Panacea also adopts the same sliding window strategy. Even though some other methods support generating long videos by producing frames 1 to N, then N+1 to 2N, and so on, our Glad approach offers more flexibility and does not determine the hyperparameters (such as overlap length, condition length, etc.) when generating long video. It is particularly advantageous when only a few frames need to be generated or when input conditions suddenly change. **As proved by reviewer XvyP, our work was finished in May 2024.** The work Delphi mentioned was posted on arXiv one month later, which should be considered as contemporaneous work with ours.
>
> Regarding the unseen scenes claimed in our paper, conventional long video generation methods require complete future trajectory information in advance, which is impractical in real-world scenarios. When an ego vehicle deviates from its planned trajectory during driving, the new trajectory becomes "unseen" to the model. **In other words, these scenes are considered "unseen" because they emerge in future timesteps rather than a predefined offline new path.** While existing long video generation methods can handle offline new trajectories, they lack the flexibility of our frame-by-frame generation paradigm in adapting to temporal uncertainties. Our approach excels at handling dynamic trajectory changes since it doesn't require pre-planning the entire sequence and can adjust to new trajectory inputs immediately.
> ***
> **W2: On the technical novelty concerns.**
>
> **A2:** We are glad to discuss with you the difference between other methods and our Glad. We think the frame-by-frame video generation ability comes from the paradigm difference. Let's explain the reason.
> First, for generating video frame-by-frame, both Drive-WM and Panacea have a limitation. For video generation, these models must output at least 2 frames and keep the last one. In contrast, Glad only needs to generate 1 frame everytime. On the other hand, temporal-layer based video generation models (such as Panacea) have specific training requirements. In the training stage, they must learn how to model long temporal sequences. This means that the length of training video clips is important and heavy computation is required. The Streaming Data Sampler is concise and intuitive, but it is a new training paradigm. It is designed for frame-by-frame video generation. Not only that, the sampler is also efficient for alleviating error accumulation. As shown in Table 5(b), this sampler can reduce the GPU memory requirement.
> ***
> **W3: On the empirical validation concerns.**
>
> **A3: We kindly remind that the object tracking results evaluated by StreamPETR have been shown in Table 2.**  Regarding the validation of long sequence videos, the current FVD only supports a maximum of 16 frames per video, so we only show the results for 16-frame videos. To address those concerns, we are pleased to provide 40-frame videos from the nuScenes validation dataset in the supplementary material. For the word "online" in our paper, it does not mean on-the-fly rendering; it represents frame-by-frame video generation. We will clarify this concept in the revised paper. The generation speed for diffusion-based video generation models is still a main bottleneck. Recent works, such as SANA[1], propose an efficient high-resolution image synthesis method. Considering that the acceleration of diffusion process is not the main focus of this paper, but due to the properties of our Glad, we can directly convert these efficient image synthesis methods for video generation.
> ***
> **W4: On the presentation concerns.**
>
> **A4:** Thanks for your remind, we have correct the mistake in the manuscript.
> ***
> We appreciate your thoughtful review and we hope to address your concerns. Please let us know if you'd like any further discussion.
> ***
> [1]Xie, Enze, et al. "SANA: Efficient High-Resolution Image Synthesis with Linear Diffusion Transformers." arXiv preprint arXiv:2410.10629 (2024).

---

> > ### Comment · Reviewer_wGv7 · 2024-11-26
> >
> > Again, your claim is to deal with “unseen or restricted length”, that to me, means your method is targeting at any length video generation. Otherwise, the design to generate frame-by-frame is only improving the efficiency. In that case, I do not see the response to my earlier comment, complexity analysis in the response.
> >
> > Also I find it funny that, the authors claimed that this work is done in May 2024 so it should not take one month later paper into any consideration. If this is a NeurIPS submission, it is not reasonable to mention a paper in June 2024. But, we are in ICLR submission round, if I may point it out… In addition, my original point is not really to directly penalize this work by citing an arxiv paper. My purpose is to emphasize some of the earlier approach can generate unseen sequence, that contradicts to your motivation. However, you choose to not directly respond, but mentioning “this paper is one month later than May 2024…” In this case, it does not clarify my concern. Your response is “DriveWM generates 2 frames and drop one, yours is generating frame by frame …” This to me seems more like an explanation the technical differences but not the motivation. Why we should do that? If there is no other benefit? Or on the other hand, how does the flexibility of your method give any benefit towards to earlier approaches?
> >
> >
> > A more interesting thing is, the reviewer you mentioned reads your NeurIPS submission, maintaining a negative score claiming limited novelty. I partially agree, on the one hand, the technical novelty is marginal but true. On the other hand, I find these claims are not throughly examined and not shown significant improvement over the previous works.

---

> > > ### Author Response · Authors · 2024-11-28
> > > **Follow-up Official Comments by Authors**
> > >
> > > Dear Reviewer,
> > >
> > > We apologize for any confusion in our previous response. As we have explained the "unseen or restricted length" issue in A1 of our last reply, we first addressed the "restricted length" issue and then the "unseen" aspect in the second paragraph. We appreciate your time and would be grateful if you could review A1 of our last response for clarity.
> > >
> > > Regarding the mention of when this work was completed, we were simply stating the relevance of our work, not intending to disregard subsequent studies. We apologize for any misunderstanding that may have arisen. We hope to address your concerns. Please let us know if you'd like any further discussion.
> > >
> > > Best regards.

---

> > > > ### Comment · Reviewer_wGv7 · 2024-12-02
> > > > **Maintaining my score.**
> > > >
> > > > Thanks for your reply. In my humble opinion, can you provide some empirical evidence regarding the benefits? Your point saying earlier methods might need some extra hyper-parameters or cannot respond to some immediates shift of the planning results, yet the downstream tasks are done in 3D object detection.
> > > >
> > > > It has been a while that I did not catch upon your paper. Could you give more concrete evidence to validate your claim?
> > > >
> > > > In short, I think a better way to prove your method is better than others is providing more evidence, this also applies to my reply in another thread. Given the fact that the domain of autonomous driving world model is growing rapidly, it might be hard for the researchers in this field but there is not much what we can do.
> > > >
> > > > Another interesting fact that, after reading the revised PDF, I found the introduction was not revised at all. It means that if I read the paper again, I will still find it confusing. This makes it hard to convince me the authors will take our suggestions seriouesly and revise their manuscript accordingly.
> > > >
> > > > In their revised experiment section, what they include is:
> > > >
> > > > - Table 2. 3D object detection task, compared with Panacea, DriveWM.
> > > > - Table 2. 3D object multiple tracking task, compared with Oracle only, no baseline at all.
> > > > - Table 3. Data synthesis task, compared with Panacea in setting without nuImage, showing the same performance
> > > > - Table 4. HD map construction, compared with custom baseline only.
> > > >
> > > > The only experiment, that might reflect the benefit the authors tried to claim,
> > > >
> > > > > "In other words, these scenes are considered "unseen" because they emerge in future timesteps rather than a predefined offline new path. While existing long video generation methods can handle offline new trajectories, they lack the flexibility of our frame-by-frame generation paradigm in adapting to temporal uncertainties. Our approach excels at handling dynamic trajectory changes since it doesn't require pre-planning the entire sequence and can adjust to new trajectory inputs immediately."
> > > >
> > > > is the tracking task. But if I understand correctly, in order to test this in nuScenes, you are using the original scene instead of the generated new trajectory right? How can you claim this with the current experimental setting?
> > > >
> > > > In addition, recent trend of world models are tested on end-to-end autnomous driving which might be a better battlefield to verify the temporal consistency of the generated scenes. Yet, this paper lacks any of these to show case their effectiveness.
> > > >
> > > > Another thing is the revised PDF still lacks of complexity analysis. From table 1 to Table 4, there is not a run time comparison with earlier method, yet the authors tried to claim their approach to generating frame-by-frame is more efficient than other generating 1-N frames together.
> > > >
> > > > In the end, even though ICLR grant extra revision time for the authors to include these, I find the main paper still does not include any of these. So I am sorry but to conclude that this current submission is still not yet for a top-tier conference and wish the authors to revise their paper and resubmit to a future venue.

---

> > > > > ### Author Response · Authors · 2024-12-03
> > > > > **Follow-up II Official Comments by Authors**
> > > > >
> > > > > Dear Reviewer,
> > > > >
> > > > > We regret that our response has not met your expectations. We appreciate your time and effort to review our work.
> > > > >
> > > > > Best regards.

---

### Official Review · Reviewer_8mSK · 2024-11-04

**Soundness:** 3
**Presentation:** 2
**Contribution:** 3
**Rating:** 8
**Confidence:** 5

**Summary:**

The authors propose a latent propagation method for generating long and consistent driving video in a frame by frame manner.
The latent pronation module simply use the denoised latent of frame t as the noise latent for denoting the latent of frame t+1.
A special streaming data sampler is designed to enable the training of this model.
Extensive experiments on nuscenes showcases the generation quality and effectiveness of using generated data to improve the downstream perception tasks.

**Strengths:**

1. The authors propose a simple yet effective framework for video generation in driving scenario, the idea of latent propagation is interesting.
2. Although missing some comparison, the experiments and ablation studies are generally solid.

**Weaknesses:**

1. Confusing writing and missing key reference. I think the whole idea of Glad is to denoise frame t+1 from the fully denoised latent of frame t instead of gaussian noise as suggested in Figure 2, but the whole section 4 tells very little about this fact except the eq 3. Also, the idea is not utterly new as this is know as image noise prior in TRIP[1], which also show great FID/FVD performance compared with other video generation models.
2. Serious issue in paper organization. The section 3 is pretty much redundant as the latent diffusion is introduced in almost 3 years ago and is now well known to the whole AI community and general technique enthusiasts. Also there are tens of works applying stable diffusion or stable video diffusion for video forecasting or world modeling in autonomous driving.
3. Poor literature review. Why not compare with DriveWM[2] and MagicDrive[3] which also do controlled multi-view multi-frame generation and report results on detection and mapping. Also if the authors want to highlight the streaming generation in driving scene(as indicated in the paper title), Vista[4] is a very important method to compare.

[1] TRIP: Temporal Residual Learning with Image Noise Prior for Image-to-Video Diffusion Models, CVPR 2024
[2] Driving into the Future: Multiview Visual Forecasting and Planning with World Model for Autonomous Driving, CVPR 2024
[3] MAGICDRIVE: STREET VIEW GENERATION WITH DIVERSE 3D GEOMETRY CONTROL, ICLR 2024
[4] Vista: A Generalizable Driving World Model with High Fidelity and Versatile Controllability, NeurIPS 2024

**Questions:**

1. Intuitively, using the denoised previous frame latent as noise latent seems poses heavy restriction on drastic content change during generation. Could the authors provide more insight about how Glad balance between  frame-to-frame consistency and the quick motion of generated video.

---

> ### Author Response · Authors · 2024-11-23
> **Official Comment by Authors**
>
> We thank reviewer 8mSK for the valuable time and constructive feedback.
> ***
> **W1: Confusing writing.**
>
> **A1:** We apologize for the confusing writing. Your summary is precise. We have rewritten method Section to highlight the main idea in this work and updated the manuscript. We would be pleased if you could read our paper again and propose some suggestions.
> ***
> **W2: Missing key reference.**
>
> **A2:** We greatly appreciate your recommendation of TRIP[1]. It is an important exploration of noise prior for video generation, and we have refered to it in the manuscript.
>
> Although both TRIP and Glad employ the term "noise prior," there are significant differences between the two methods. Glad directly uses the latent features from the previous frame as the noise prior, effectively replacing the Gaussian noise in the diffusion forward process. **In TRIP, the noise prior is more accurately described as "reference noise."** By obtaining this reference noise, the model only needs to predict the deviation from it in the backward process. The prior approach in TRIP is similar to the "anchor point prior" used in 2D detection.
>
> Furthermore, TRIP predicts the noise prior for the entire video clip from the initial frame, which can be inaccurate over large time spans. In contrast, Glad passes the noise prior frame by frame, which is more reasonable.
> ***
> **W3: Serious issue in paper organization.**
>
> **A3:** Thank you for your suggestion. Section 3 is somewhat redundant and, as you previously mentioned, it has resulted in a lack of focus in the current structure.  Therefore, we move Section 3 to the Appendix.
> ***
> **W4: Poor literature review.**
>
> **A4:** Thank you for your reminders. We have added DriveWM's results to Table 2 in the manuscript.
>
> MagicDrive is a single-frame method, and its perception performance is assessed using the single-frame model BEVFusion. Therefore, it is somewhat unfair to compare the detection performance to Glad in Table 2, given that it is considerably lower. By the way, we have updated the FVD result of MagicDrive in Table 1.
>
> As reviewer XvyP has noted, this paper was drafted in May, making Vista a contemporary work of ours. The settings for FID and FVD are different from ours, as Vista crops the images to $224 \times 224$ while we use $512 \times 256$. However, considering the relevance of Vista to our research, we have also included discussion of it within the manuscript.
> ***
> **Q1: Balancing consistency and motion in generated video.**
>
> **A5:** We are glad to discuss the balance between frame-to-frame consistency and quick motion with you. In the field of video generation, consistency is vitally important. Current evaluation metrics such as FVD also mainly focus on consistency. On the contrary, quick motion is important for practical application value. Quick motion issue primarily appear in the foreground objects of generated videos. During rapid movement, objects may appear to jump between frames. We think the object-centric evaluation metrics like MOTA might help measure the consistency of these objects across frames.
>
> To maintain consistency in this work, for simplicity, we only make an effort through the input condition. Specifically, we keep the unique color of each object's bounding box consistent throughout a video. Despite its simplicity, our method achieves comparable AMOTA performance to approaches that directly inject subject IDs, such as SubjectDrive[2]. Quick motion remains a significant challenge in video generation, presenting substantial opportunities for future improvements.
> | Method | AMOTA↑ | AMOTP↓ |
> | :---: | :---: | :---: |
> | SubjectDrive | 23.4 | 1.544 |
> | Glad | 22.7 | 1.526 |
>
> ***
> We appreciate your thoughtful review and we hope to address your concerns. Please let us know if you'd like any further discussion.
> ***
> [1]Zhang, Zhongwei, et al. "TRIP: Temporal Residual Learning with Image Noise Prior for Image-to-Video Diffusion Models." CVPR. 2024.
>
> [2]Huang, Binyuan, et al. "Subjectdrive: Scaling generative data in autonomous driving via subject control." arXiv preprint arXiv:2403.19438 (2024).

---

> > ### Comment · Reviewer_8mSK · 2024-11-29
> >
> > The response generally addressed my main concerns.
> > But I found some subtle details in the newly added text about training.
> > Glad uses multi-scale training for both the generation and downstream and as far as I know this is not a general practice for early works like DriveWM and Vista.
> > So I wonder how much performance improvement does this training trick account for in Table 1, 2, 3.
> >
> > This severely influence my judgement about the true strength of the streaming generation setting.

---

> > > ### Author Response · Authors · 2024-11-30
> > > **Follow-up Official Comments by Authors**
> > >
> > > Dear Reviewer,
> > >
> > > Thank you for highlighting the multi-scale training configuration. We constructed our dataset with reference to Panacea's dataset, which uses multi-scale settings by default. After your reminder, we realized that DriveWM and Vista did not employ multi-scale training. This raised our serious concern, so we retrained our generative model with a fixed scale factor of 0.32 and tested it in downstream tasks. As the table shows, multi-scale training has minimal impact on both perception and generation performance. We suspect that multi-scale training in Panacea may primarily serve to align with StreamPETR's data processing pipeline.
> > >
> > > | Mult-scale Training | FID$\downarrow$ | FVD$\downarrow$ | mAP$\uparrow$ | NDS$\uparrow$ | AMOTA$\uparrow$ | AMOTP$\downarrow$ |
> > > |:----:|:----:|:----:|:----:|:----:|:-------:|:-------:|
> > > | | 12.78 | 211 | 27.9 | 41.2 | 22.4 | 1.529 |
> > > | $\checkmark$ | 12.57| 207 | 28.3 | 41.3 | 22.7 | 1.526 |
> > >
> > > However, using multi-scale training creates an inconsistency with existing methods like DriveWM and Vista. We apologize for these setting discrepancies relative to previous works. We will include this ablation study in the paper's appendix and specify in the main experimental table which methods used multi-scale versus single-scale settings. We hope to address your concerns. Please let us know if you'd like any further discussion.
> > >
> > > Best regards.

---

> > > > ### Comment · Reviewer_8mSK · 2024-11-30
> > > >
> > > > Thanks for the timely reply and I have no further questions.

---

### Official Review · Reviewer_B154 · 2024-11-05

**Soundness:** 3
**Presentation:** 3
**Contribution:** 2
**Rating:** 6
**Confidence:** 5

**Summary:**

Collecting large-scale real-world driving data is expensive and labor-intensive. Diffusion generative models have achieved remarkable success in creating diverse and  high-quality video  from textual prompts. Therefore, this paper proposes to use diffusion model as a data generator which produce video data frame by frame in an online manner. To maintain temporal consistency, the authors propose latent variable propagation. The Glad can be used as generator and simulator, according to input. Experimental results show Glad is able to to generate videos of arbitrary lengths and exhibiting good flexibility in the variations of simulated trajectory.

**Strengths:**

1. The motivation for using diffusion models for data generation in autonomous driving is clearly a promising direction, reducing the labor-intensity.

2. The paper is well-written and easy to follow. Fig.2 provides a clear illustration of the video data generation pipeline, which makes me easy to understand the method proposed in this paper.

3. The framework is designed similar to the recurrent network, using the previous frame to predict the current frame. This is make sense for achieving arbitrary length video generation.

**Weaknesses:**

1. From the method and evaluation sections of the paper, it's not very clear whether this method is able to address the error accumulation when the frame is generated one by one. In the method section, it will be useful to clarify this and explain how this method is able to do so.

2. It is unclear to me how many training data used for Glad and other methods in Table 1, such as Panacea and Oracle.

3. It would be interesting if the authors could show the detailed detection performance of 10 classes on the nuScenes.

**Questions:**

1. Can the authors clarify with why Glad cannot achieve further improvement by using more video frames in Table 2? I would appreciate if this was clarified.

2. How about the performance of Glad just use the generated data for training.

3. Effective generation of corner cases is essential for providing key data support the model, as mentioned in abstract. I'm wonder if Glab is able to generate such data.

---

> ### Author Response · Authors · 2024-11-23
> **Rebuttal 1/2**
>
> We thank reviewer B154 for the valuable time and constructive feedback.
> ***
> **W1: The error accumulation issue of Glad.**
>
> **A1:**  Error accumulation is a significant challenge in video generation, particularly for long video sequences.  We also faced this issue at the outset, but ultimately resolved it by constructing **an appropriate noise prior**. The analysis and resolution process are as follows:
>
> As shown in Fig. 8(a) in the Appendix, **in our initial approach, we constructed the current noise prior by extracting the latent features from the real image of previous frame.** Under this strategy, the model was trained solely on the transition between two consecutive frames. During testing, errors accumulated gradually with each additional frame, ultimately leading to a collapse in image quality.
>
> To address this issue, as depicted in Fig. 8(b), **we introduced noise into the latent features from previous real frame.** This strategy was designed to encourage the model to reconstruct the current frame from features that included errors. However, the noise we introduced not align with the actual patterns of errors observed during testing. Consequently, the model not achieve the desired improvement.
>
> Building on these attempts, we have adopted the current construction strategy, as shown in Fig. 8(c) in Appendix. Instead of directly using the real features from the previous frame to construct the noise prior, **we employ the denoised previous features.** On one hand, the denoised features contain errors compared to the features extracted from the real image. On the other hand, the model is still required to accurately reconstruct the current image during training. This interplay has endowed the model with the capability to recover the current frame from erroneous inputs. By extending the training frames, the error propagation in training more closely mirrors that of the testing phase. Ultimately, it enhances the model's ability to correct accumulated errors.
>
> In the Table below, compared to other strategies, the noise prior $\mathbf{z}^{n-1}_{denoised}$ achieves the best performance in all evaluation metrics. Therefore, in our Glad approach, we utilize the denoised latent feature.
> | Strategy | Noise prior | FID$\downarrow$| FVD$\downarrow$| mAP$\uparrow$| NDS$\uparrow$|
> | -------- | ----------- | ----- | ----- | ---- | ---- |
> | (a) | $\mathbf{z}^{n-1}_{real}$ | 49.02 | 607 | 15.5 | 30.9 |
> | (b) | $\mathbf{z}^{n-1}_{noised}$ | 32.43 | 362 | 19.5 | 35.5 |
> | (c) | $\mathbf{z}^{n-1}_{denoised}$ | **11.18** | **188** | **28.3** | **41.3** |
> ***
> **W2: About the training data.** It is unclear to me how many training data used for Glad and other methods in Table 1, such as Panacea and Oracle.
>
> **A2:**  For the training of our generative model, Glad used the same training dataset as Panacea. The term "Oracle" is not a generative algorithm. We apologize for any misunderstanding and will provide an explanation for it.
>
> In Table 2, we evaluate the authenticity of our generated data by employing a pre-trained StreamPETR on the validation set. **The "Oracle" row represents the results of StreamPETR when applied to the real nuScenes validation set.** The other rows indicate the performance of StreamPETR on synthetic validation set. The closer the perceptual results to the "Oracle," the higher quality of the generated data.
> ***
> **W3: Detail of the detection performance.** It would be interesting if the authors could show the detailed detection performance of 10 classes on the nuScenes.
>
> **A3:** Certainly, the detailed detection performance of the 10 classes can reflect the generation quality of each category. Since Panacea does not provide results broken down by category, we compare our results with the Oracle results from StreamPETR. It can be observed that common categories such as Car and Traffic cone have higher generation quality compared to rare categories like Construction Vehicles and Trailer. The "CV" and "TC" are the abbreviation of the Construction Vehicle and Traffic Cone.
>
> | | Car | Truck | Bus | Trailer | CV | Pedestrian | Motorcycle | Bicycle | TC | Barrier | mAP |
> | :---: | :---: | :---: | :---: | :---: | :---: | :---: | :---: | :---: | :---: | :---: | :---: |
> | Oracle | 0.585 | 0.333 | 0.308 | 0.115 | 0.115 | 0.445 | 0.389 | 0.523 | 0.590 | 0.544 | 0.378 |
> | Glad | 0.423 | 0.219 | 0.201 | 0.061 | 0.071 | 0.306 | 0.269 | 0.292 | 0.496 | 0.495 | 0.283 |
> | **Score** | **72.3%** | **65.8%** | **65.3%** | **53.0%** | **61.7%** | **68.8%** | **69.2%** | **55.8%** | **84.0%** | **91.0%** | **74.9%** |
>
> ***

---

> ### Author Response · Authors · 2024-11-23
> **Rebuttal 2/2**
>
> **Q1: Lack of performance improvement when using more video frames.**
>
> **A4:** This is an insightful question, and we also noticed the performance saturation at 8 frames in the ablation study (see their Table 5) of StreamPETR. We have explored the reasons for this phenomenon and **attribute our preliminary conclusions to the inherent properties of the nuScenes dataset.**
>
> As we know, current controllable video generation models are mainly object-centric, focusing on objects with annotations like 3D bounding boxes. We analyzed the object consistency in nuScenes, and the results are shown in the **Fig.7 in Appendix.** As we can see, almost 96.2\% of the objects are persist for less than 8 frames, while only 3.5\% are persist between 8 frames and 16 frames. These results mean that in a generated video of 16 frames, the objects in the last frames are entirely different from those in the first frame, which leads to performance degradation.
> ***
> **Q2: Performance on generated data.**
>
> **A5:** We have examined our previous experiments and present the results in the Table below. When trained solely on synthetic data, Glad outperformed Panacea by 4.6\% mAP and 4.7\% NDS.
> | | nuImage | Generated | mAP↑ | NDS↑ | AMOTA↑ | AMOTP↓ |
> | :---: | :---: | :---: | :---: | :---: | :---: | :---: |
> | Panacea | | ✓ | 22.5 | 36.1 | - | - |
> | Glad | | ✓ | 27.1 | 40.8 | 21.6 | 1.569 |
> | Glad | ✓ | ✓ | 30.0 | 42.8 | 24.9 | 1.500 |
>
> ***
> **Q3: Generation in corner cases.**
>
> **A6:** To generate corner cases, we utilized the CODA[1] dataset, which is specifically designed for real-world road corner cases. CODA encompasses three major autonomous driving datasets, including 134 scenes from nuScenes. We focused on this subset of 134 nuScenes scenes. After filtering to identify which scenes belonged to the validation set, we retained 30 valid scenes. These 30 scenes serve as our corner case examples, and we present selected generation results in **Fig.9 in Appendix.** We showcase five corner cases that demonstrate our generation capabilities: a car making a sudden turn in the opposite lane (case 1), traffic cones along the roadside (cases 2 and 3), unusual roadside obstacles (case 4), and unconventional road signs (case 5).
> ***
> We appreciate your thoughtful review and we hope to address your concerns. Please let us know if you'd like any further discussion.
>
> [1] Li, Kaican, et al. "Coda: A real-world road corner case dataset for object detection in autonomous driving." European Conference on Computer Vision. Cham: Springer Nature Switzerland, 2022.

---

> > ### Comment · Reviewer_B154 · 2024-11-27
> >
> > Thanks for the detailed responses. I will maintain my previous review score.

---

### Author Response · Authors · 2024-11-23
**General Response**

We sincerely thank all reviewers for their valuable and constructive comments. We have tried our best to revise the paper to address all concerns. *All revisions are marked in  **blue.*** Specifically:
***
- **@Reviewer B154**, we have analyzed the influence of noise prior on error accumulation, as shown in Fig.8 in the appendix. The experimental results in Tab. 6 demonstrate that using generated latent features as noise prior achieves the best performance across all evaluation metrics.

- **@Reviewer B154**, we have investigated why using additional video frames does not improve performance. Our analysis of object persistence in frames, shown in Fig.7 in the Appendix, reveals that this limitation is due to the inherent characteristics of the nuScenes dataset.

- **@Reviewer B154**, **@Reviewer wGv7** and **@Reviewer MnsB** ,we have included the generation results for corner case scenes in Fig.9 in the Appendix and provided additional long-form videos at the supplementary material.

- **@Reviewer 8mSK**, we have moved the original Section 3 to the Appendix section and have made the method section more clear.

- **@Reviewer 8mSK** and **@Reviewer XvyP**, we have included the results from contemporary video generation methods Drive-WM[1] and MagicDrive[2] in Tab.2.

- **@Reviewer MnsB**, we have incorporated a temporal layer into our framework for an ablation study. %while maintaining the original Gaussian noise as prior.
The results are shown in Tab.7 in the Appendix. Compared to using the temporal layer, our Glad achieves a better balance between spatial modeling and temporal propagation.

- **@Reviewer wGv7**, we have replaced all "cite" commands with "citep".

- **@Reviewer MnsB**, to further explore the simulator potential, we have included a video in the supplementary material, demonstrating the end-to-end autonomous driving model VAD[3] performing inference on generated nuScenes validation set by our Glad.

Please let us know if you have any further questions. We are always looking forward to open discussions.

Sincerely,

Authors
***
[1]Wang, Yuqi, et al. "Driving into the future: Multiview visual forecasting and planning with world model for autonomous driving." Proceedings of the IEEE/CVF Conference on Computer Vision and Pattern Recognition. 2024.

[2]Gao, Ruiyuan, et al. "Magicdrive: Street view generation with diverse 3d geometry control." arXiv preprint arXiv:2310.02601 (2023).

[3]Jiang, Bo, et al. "Vad: Vectorized scene representation for efficient autonomous driving." Proceedings of the IEEE/CVF International Conference on Computer Vision. 2023.

---

### Author Response · Authors · 2024-11-23
**PDF Revision And Supplementary Material Update**

Dear Reviewers,

We appreciate your feedback and have revised our manuscript accordingly:

- Added more related works, such as TRIP and Vista. (Pg 3)
- Moved the original section 3 (preliminary) to Appendix A.1 (Pg 15)
- Revised method section to focus on our framework and clarify our training details (Pg 4-6)
- Added comparison with MagicDrive and Drive-WM in experiments (Pg 7)
- Fixed the format of the citations
- Added more experiments and visualization results in Appendix A.2 (Pg 15–17)
- Added 40-frame long videos and VAD simulation videos in supplementary material

---

We hope the newly provided results alleviate any confusion. We welcome any further feedback you may have and look forward to the opportunity to continue refining our work.

---

### Public Comment · ~Jiawei_Zhou4 · 2025-09-28

Why is the temporal consistency of the VAD simulation in the supplementary materials so poor?

---

### Meta-Review · Area_Chair_3Yz2 · 2024-12-21

**Metareview:**

This paper got very diverging scores, so it takes some time to read through all comments and discussions. This paper presents Glad, a novel framework for online generation of multi-view video data for autonomous driving simulation. While reviewers acknowledge some limitations, most reviewers ultimately find the paper's contributions to be promising and worthy of acceptance.
As for some issues raised by Reviewer wGv7 and XvyP, authors have given reasonable explanations in details. To summarize,
1. The proposed frame-by-frame generation pipeline provides more flexibility in autonomous driving simulation.
2. Latent variable propagation is different from cross-frame attention, noise inversion in training-free video generation works and noise prior in PYoCo and VideoFusion.
3. DriveDreamer-2 is trained and evaluated on the privately annotated videos, 12Hz instead of 2Hz annotation in original nuScenes dataset, which leads to better FVD scores.
4. Delphi is a concurrent work which does not contradicts to the submitted work.

**Additional Comments On Reviewer Discussion:**

Each reviewer actively participate in the rounds of discussion with authors, esp. Reviewer wGv7 who holds negative opinion. Authors have responded in details to explain the concerns of reviewers, including motivation, novelty and comparison with sota methods.

---

### Decision · Program_Chairs · 2025-01-22

Accept (Poster)